# Linear reinforcement learning in planning, grid fields, and cognitive control

Payam Piray 🄳 [1✉] & Nathaniel D. Daw 🄳 [1]

It is thought that the brain's judicious reuse of previous computation underlies our ability to plan flexibly, but also that inappropriate reuse gives rise to inflexibilities like habits and compulsion. Yet we lack a complete, realistic account of either. Building on control engineering, here we introduce a model for decision making in the brain that reuses a temporally abstracted map of future events to enable biologically-realistic, flexible choice at the expense of specific, quantifiable biases. It replaces the classic nonlinear, model-based optimization with a linear approximation that softly maximizes around (and is weakly biased toward) a default policy. This solution demonstrates connections between seemingly disparate phenomena across behavioral neuroscience, notably flexible replanning with biases and cognitive control. It also provides insight into how the brain can represent maps of long-distance contingencies stably and componentially, as in entorhinal response fields, and exploit them to guide choice even under changing goals.

[1] Princeton Neuroscience Institute, Princeton University, Princeton, NJ, USA. ✉email: ppiray@princeton.edu

The brain exhibits a remarkable range of flexibility and inflexibility. A key insight from reinforcement learning (RL) models is that humans' ability flexibly to plan new actions—and also our failures sometimes to do so in healthy habits and disorders of compulsion—can be understood in terms of the brain's ability to reuse previous computations[1–5]. Exhaustive, "model-based" computation of action values is time-consuming; thus, it is deployed only selectively (such as early in learning a new task), and when possible, the brain instead bases choices on previously learned ("cached," "model-free") decision variables[1,4]. This strategy saves computation, but gives rise to slips of action when cached values are out-of-date.

However, while the basic concept of adaptive recomputation seems promising, this class of models—even augmented with refinements such as prioritized replay, partial evaluation, and the successor representation—has so far failed fully to account either for the brain's flexibility or its inflexibility[5,6]. For flexibility, we still lack a tractable and neurally plausible account how the brain accomplishes the behaviors associated with model-based planning. Conversely, the reuse of completely formed action preferences can explain extreme examples of habits (such as a rat persistently working for food it does not want), but fails fully to explain how and when these tendencies can be overridden, and also many subtler, graded or probabilistic response tendencies, such as Pavlovian biases or varying error rates, in cognitive control tasks.

Here, we introduce a new model that more nimbly reuses precursors of decision variables, so as to enable a flexible, tractable approximation to planning that is also characterized by specific, graded biases. The model's flexibility and inflexibility (and its ability to explain a number of other hitherto separate issues in decision neuroscience) are all rooted in a new approach to a core issue in choice. In particular, we argue that the central computational challenge in sequential decision tasks is that the optimal decision at every timepoint depends on the optimal decision at the next timepoint, and so on. In a maze, for instance, the value of going left or right now depends on which turn you make at the subsequent junction, and similarly thereafter; so, figuring out what is the best action now requires, simultaneously, also figuring out what are the best choices at all possible steps down the line. This interdependence between actions is a consequence of the objective of maximizing cumulative expected reward in this setting and is reflected in the Bellman equation for the optimal values[7]. However, it also greatly complicates planning, replanning, task transfer, and temporal abstraction in both artificial intelligence and biological settings[8].

How, then, can the brain produce flexible behavior? Humans and animals can solve certain replanning tasks, such as reward devaluation and shortcuts, which require generating new action plans on the fly[6,9–12]. It has been argued that the brain does so by some variant of model-based planning; that is, solving the Bellman equation directly by iterative search[1,4]. However, we lack a biologically realistic account how this is implemented in the brain[3]; indeed, because of the interdependence of optimal actions, exhaustive search (e.g., implemented by neural replay[13]) seems infeasible for most real-world tasks due to the exponentially growing number of future actions that must each be, iteratively and nonlinearly optimized. It has thus also been suggested that the brain employs various shortcuts that rely on reusing previously computed ("cached") quantities, notably model-free long-run values[14,15].

One such proposal, which is perhaps the most promising step toward a neurally realistic planning algorithm is the successor representation (SR)[16], which by leveraging cached expectations about which states will be visited in future, can efficiently solve a subset of tasks traditionally associated with model-based planning[5,6]. However, it simply assumes away the key interdependent optimization problem by evaluating actions under a fixed choice policy (implied by the stored state expectancies) for future steps. This policy dependence makes the model incapable of explaining how the brain can solve other replanning tasks, in which manipulations also affect future choices[5,17]. In general, the precomputed information stored by the SR is only useful for replanning when the newly replanned policy remains similar to the old one. For instance, a change in goals implies a new optimal policy that visits a different set of states, and a different SR is then required to compute it. This is just one instance of a general problem that plagues attempts to simplify planning by temporal abstraction (e.g., chunking steps[18,19]), again due to the interdependence of optimal actions: if my goals change, the optimal action at future steps (and, hence, the relevant chunked long-run trajectories) likely also change.

Here, we adopt and build on recent advances in the field of control engineering[20–22] to propose a new model for decision-making in the brain that can efficiently solve for an approximation to the optimal policy, jointly across all choices at once. It does so by relying on a precomputed, temporally abstract map of long-run state expectancies similar to the SR, but one which is, crucially, stable and useful even under changes in the current goals and the decision policy they imply. We term the model linear RL, because it is based on Todorov's work with a class of linearly solvable Markov decision processes (MDPs)[20–22]. It provides a common framework for understanding different aspects of animals' cognitive abilities, particularly flexible planning and replanning using these temporally abstract maps, but also biases in cognitive control and Pavlovian influences on decision-making, which arise directly from the strategy of reuse.

The model is based on a reformulation of the classical decision problem, which makes "soft" assumptions about the future policy (in the form of a stochastic action distribution), and introduces an additional cost for decision policies which deviate from this baseline. This can be viewed as an approximation to the classic problem, where soft, cost-dependent optimization around a baseline, which we hereafter call the default policy, stands in for exact optimization of the action at each successor state. This enables the model efficiently to deal with the interdependent optimization problem. Crucially, the form of the costs allows the modified value function to be solved analytically using inexpensive and biologically plausible linear operations. In particular, the optimal value of any state under any set of goals depends on a weighted average of the long-run occupancies of future states that are expected under the default policy. Therefore, we propose that the brain stores a map of these long-run state expectancies across all states (the default representation, or DR), which gives a metric of closeness of states under the default policy. Because the DR depends only on the default policy, and can be used to compute a new optimal policy for arbitrary goals, the model can solve a large class of replanning tasks, including ones that defeat the SR.

Our modeling approach also addresses a number of seemingly distinct questions. First, the stability of the DR across tasks makes it a candidate for understanding the role in decision making of multiscale, temporally abstract representations in the brain, notably grid cells in the medial entorhinal cortex. These cells show regular grid-like firing patterns over space, at a range of frequencies, and have been argued to represent something akin to a Fourier-domain map of task space (e.g., the eigenvectors of the SR, equivalent to the graph Laplacian[2,23]), and could provide some sort of mechanism for spatial[24] and mental navigation[12,25,26]. However, it has been unclear how this and similar long-run temporal abstractions are actually useful for planning or navigation, because as mentioned long-run (low-frequency) expectancies over task space are not stable across tasks

due to the interdependence of policy, goals, and trajectories[27,28]. For instance, because the SR only predicts accurately under the training policy, to be even marginally useful for replanning the SR theory predicts grid fields must continually change to reflect updated successor state predictions as the animal's choice policy evolves, which is inconsistent with evidence[29–31]. The linear RL theory clarifies how the DR, a stable and globally useful long-run map under a fixed default policy, can serve flexible planning. Our theory also provides a new account for updating maps in situations which actually do require modification—notably, the introduction of barriers. We show how these give rise to additional, separable basis functions in the corresponding DR, which we associate with a distinct class of entorhinal response fields, the border cells. This aspect of the work goes some way toward delivering on the promise of such response as part of a reusable, componential code for cognitive maps[12,25].

Finally, linear RL addresses the flip side of how the brain can be so flexible: why, in some cases it is inflexible. We suggest that this is simply another aspect of the same mechanisms used to enable flexible planning. While it has long been suggested that fully model-free learning in the brain might account for extreme cases of goal-inconsistent habits (e.g., animals persistently working for food when not hungry[1]), there are many other phenomena which appear as more graded or occasional biases, such as Stroop effects[32,33], Pavlovian tendencies[34,35], slips of action[36], and more sporadic failures of participants to solve replanning tasks[6]. The default policy and cost term introduced to make linear RL tractable offers a natural explanation for these tendencies, quantifies in units of common-currency reward how costly it is to overcome them in different circumstances, and offers a rationale and explanation for a classic problem in cognitive control: the source of the apparent costs of "control-demanding" actions.

Despite its simplicity, the linear RL model accounts for a diverse range of problems across different areas of behavioral neuroscience. In the reminder of this article, we present a series of simulation experiments that demonstrate that the theory provides (i) a biologically realistic, efficient and flexible account of decision making; (ii) a computational understanding of entorhinal grid and border cells that explains their role in flexible planning, navigation, and inference; (iii) a soft and graded notion of response biases and habits; (iv) an understanding of cognitive control that naturally links it to other aspects of decision systems; and (iv) a normative understanding of Pavlovian-instrumental transfer (PIT).

## Results

**The model.** In Markov decision tasks, like mazes or video games, the agent visits a series of states $s$, and at each they receive some reward or punishment $r$ and choose among a set of available actions $a$, which then affects which state they visit next[8]. The objective in this setting is typically to maximize the expected sum of future rewards, called the "value" function. Formally, the optimal value $\bar{v}^*$ of some state is given by the sum of future rewards, as a series of nested expectations:

$$\bar{v}^*(s_t) = r(s_t) + \max_{a_t} \sum_{s_{t+1}} P(s_{t+1}|s_t, a_t)[r(s_{t+1}) + \max_{a_{t+1}} \sum_{s_{t+2}} P(s_{t+2}|s_{t+1}, a_{t+1})[r(s_{t+2}) + \cdots]], \quad (1)$$

or equivalently in recursive form by the Bellman equation[7]:

$$\bar{v}^*(s_t) = r(s_t) + \max_{a_t} \sum_{s_{t+1}} P(s_{t+1}|s_t, a_t)\bar{v}^*(s_{t+1}), \quad (2)$$

where $s_t$, $r_t$, and $a_t$ denote the state, reward, and action at time $t$, respectively. Across all states, this results in a set of inter-dependent optimization problems, which can be solved, for

instance, by iterative search through the tree of future states, computing the maximizing action at each step[8]. However, in realistic tasks with large state spaces, this iterative, nonlinear computation may be intractable.

Note that prediction can be used for action choice or computing an action selection policy: once we have computed $\bar{v}^*$ (the optimal future reward available from each state), we can just compare it across actions to find the best action in any particular state and form a policy, $\pi^*$; for instance, we can evaluate the max in Eq. (2) for any state, plugging in the optimal values of successor states without further iteration. However, note also that this depends on having already found the maximizing action at other states down the line, since $\bar{v}^*$ depends, recursively, on which actions are taken later, and this in turn depends on the assignment of rewards to states (e.g., the agent's goals).

If we instead assumed that we were going to follow some given, not necessarily optimal, action selection policy $\pi$ at each subsequent state (say, choosing randomly), then Eq. (2) would be replaced by a simple set of linear equations (eliminating the nonlinear function "max" at each step) and relatively easily solvable. This observation is the basis of the SR model[2,5,6,16], which computes values as

$$\bar{\mathbf{v}}^\pi = \mathbf{S}^\pi \mathbf{r}, \quad (3)$$

where (in matrix–vector form) $\bar{\mathbf{v}}^\pi$ is a vector of long-run state values under the policy $\pi$; $\mathbf{r}$ a vector of state rewards; and $\mathbf{S}^\pi$ a matrix measuring which subsequent states one is likely to visit in the long run following a visit to any starting state: importantly, assuming that all choices are made following policy $\pi$. However, although this allows us to find the value of following policy $\pi$, this does not directly reveal how to choose optimally. For instance, plugging these values into Eq. (2) would not produce optimal choices, since $\bar{\mathbf{v}}^\pi$ (the value of choosing according to $\pi$ in the future) in general does not equal the value, $\bar{\mathbf{v}}^*$, of choosing optimally. The only way to find the latter using Eq. (3) is by iteratively re-solving the equation to repeatedly update $\pi$ and $\mathbf{S}$ until they eventually converge to $\pi^*$, i.e., the classic policy iteration algorithm.

A line of research in control engineering has shown that a change in the formulation of this problem, which we refer to as *linear RL*, greatly simplifies the Bellman equation[20–22]. In this paper, we build on this foundation to revisit questions of flexibility and inflexibility in biological learning. To derive this simplification, we first assume a one-to-one, deterministic correspondence between actions and successor states (i.e., for every state $s'$ reachable in one step from some $s$, assume there is a corresponding action $a$ for which $p(s'|s, a) = 1$, which is simply denoted by its destination, $s'$). This assumption, which we relax later, fits many problems with fully controllable, deterministic dynamics, such as spatial navigation (where for each adjacent location, there is a corresponding action taking you there). Second, linear RL seeks to optimize not a discrete choice of successor state (action), but a stochastic probability distribution $\pi$ over it[21,22]. Finally, it redefines the value function to include not just the one-step rewards $r$ but also at each step a new penalty[20–22], called a "control cost," $KL(\pi||\pi^d)$, which is increasing in the dissimilarity (Kullback–Leibler (KL) divergence) between the chosen distribution $\pi$ and some *default* distribution, $\pi^d$.

Linear RL is most naturally a formalism for modeling tasks in which there are some default dynamics (e.g., a rocket in a gravitational field) and costly actions to modify them (e.g., firing thrusters burning different amounts of fuel). Alternatively, here we view it as an approximation to the original value function, where the additional penalty terms modify the original problem

to a related one that can be more efficiently solved. This is because linear RL deals with the problem of the interdependence of the optimal actions across states[20–22]: the default policy $\pi^d$ represents a set of soft assumptions about which actions will be taken later, which are optimized into an optimal stochastic distribution $\pi^*$ that is approximately representative of the optimal (deterministic) subsequent choices in the original problem.

Efficient solution is possible because, substituting the penalized rewards into the Bellman equation, the optimal value function is now given by a non-recursive, linear equation[21,22]:

$$\exp(\mathbf{v}^*) = \mathbf{MP} \exp(\mathbf{r}), \qquad (4)$$

such as can be computed by a single layer of a simple, linear neural network. Here, $\mathbf{v}^*$ is a vector of the optimal values (now defined as maximizing cumulative reward minus control cost) for each state; $\mathbf{r}$ is a vector of rewards at a set of "terminal" states (i.e., various possible goals); $\mathbf{P}$ is a matrix containing the probability of reaching each goal state from each other, nonterminal, state; and the key matrix $\mathbf{M}$, which we call the DR, measures the closeness of each nonterminal state to each other nonterminal state (in terms of expected aggregate cost to all future visits) under the default policy. This is similar to the SR ($\mathbf{S}^\pi$, Eq. (3)), except that it is for the optimal values $\mathbf{v}^*$ (not the on-policy values $\mathbf{v}^\pi$), and $\mathbf{v}^*$ is systematically related to optimal values as defined in the original problem ($\bar{v}^*$, Eq. 1), with the difference being the additional penalties for deviation from the default policy. But these exert only a soft bias in $\pi^*$ toward $\pi^d$, which furthermore vanishes altogether in an appropriate limit (see "Methods"). Thus, while $\mathbf{M}$ does depend on the default policy $\pi^d$, it is stable over changes in goals and independent from $\pi^*$ in the sense that it can usefully find optimized policies $\pi^*$ (solving the interdependent optimization problem) even when these are far from $\pi^d$. In comparison, $\mathbf{v}^\pi$ (computed from the SR: $\mathbf{S}^\pi$) is only a useful approximation to $\mathbf{v}^*$ (and thus only helpful in finding a new $\pi^*$) when the SR's learned policy $\pi$ is near the target policy $\pi^*$. Effectively, linear RL works by introducing a smooth approximation of the "max" in Eq. (2), since the log-average-exp (with the average here taken with respect to the default distribution, $\pi^d$) of a set of values approximates the maximum. The control costs, then, simply capture the difference between the original solution and the smooth approximate one. Note that distinguishing between terminal and nonterminal states is necessary, as only for this type of finite decision problem are the optimal values linearly computable; however, this places few limits on the flexibility of the model (see "Discussion").

**Model performance**. The optimized policy in this model balances expected reward with control cost, and is generally stochastic rather than deterministic, like a softmax function (Fig. 1a, b). We evaluated the performance of linear RL as an approximation to exact solution by considering a difficult, seven-level decision tree task in which each state has two possible successors, a set of costs are assigned randomly at each state, and the goal is to find the cheapest path to the bottom. We conducted a series of simulations, comparing linear RL with a set of benchmarks: exact (model-based) solution, and a set of approximate model-based RL agents[14] that optimally evaluate the tree up to a certain depth, then "prune" the recursion at that leaf by substituting the exact average value over the remaining subtree (Fig. 1c; in the one-step case this is equivalent to the SR under the random walk policy). For linear RL, the default policy was taken as a uniform distribution over possible successor states. Except where stated explicitly, we use the same fixed uniform default policy for all simulations, so as to showcase the ability of linear RL to successfully plan without updating or relearning task-specific policy

expectations, as is generally needed for the SR. Linear RL achieved near-optimal average costs (Fig. 1d). Note that the D1 model in Fig. 1d is equivalent to the SR for the random walk policy (i.e. a uniform distribution over successor states), because it chooses actions using current reward plus the value of successor states computed based on a uniform policy.

An important aspect of linear RL is that the DR, $\mathbf{M}$, reflects the structure of the task (including the distances between all the nonterminal states under the default policy) in a way that facilitates finding the optimal values, but is independent of the goal values $\mathbf{r}$, and the resulting optimized value and policy (Fig. 2). Therefore, by computing or learning the DR once, the model is able to re-plan under any change in the value of the goals (see below) and also (with some additional computation to efficiently add an additional terminal goal state, see "Methods") plan toward any new goal with minimal further computation (Fig. 2b, c). In the case of spatial tasks, this corresponds to finding the shortest path from any state to any goal state. In fact, our simulation analysis in a maze environment revealed that linear RL efficiently finds the shortest path between every two states in the maze (Fig. 2d).

**Replanning**. In both artificial intelligence, and psychology and biology, a key test of efficient decision making is how an agent is able to transfer knowledge from one task to another. For instance, many tasks from neuroscience test whether organisms are able, without extensive retraining, to adjust their choices following a change in the rewards or goals ("revaluation," "devaluation," "latent learning") or transition map ("shortcut," "detour") of a previously learned task[6,9–11]. In general, humans and animals can successfully solve such tasks, leading to the question how, algorithmically, this is accomplished. Performance on such transfer learning tasks is particularly informative about an agent's learning strategy both because any successful adaptation exercises planning capabilities rather than trial-and-error adjustment, and also because any failures can be diagnostic of shortcuts for simplifying planning such as reuse of previously learned quantities. We explored the ability of linear RL for solving these types of replanning problems (Fig. 3).

Importantly, the model is able efficiently to solve one class of these problems that has been important in neuroscience and psychology— those involving revaluation of goals—because the DR can be used, unmodified, to solve any new problem. This corresponds to simply changing $\mathbf{r}$ in Eq. (3), and computing new values. For instance, linear RL is able to solve a version of Tolman's latent learning task (Fig. 3a), a revaluation task in which rats were first trained to forage freely in a maze with two rewarding end-boxes, but then were shocked in one of the end-boxes to reduce its value[37]. This manipulation defeats model-free RL algorithms like temporal-difference learning, because they must experience trajectories leading from the choice to the devalued box to update previously learned long-run value or policy estimates[1]. In contrast, rats are able to avoid the path leading to the devalued end-box on the first trial after revaluation, even though they had never experienced the trajectory following the devaluation[37]. Linear RL is also able to correctly update its plans using the DR computed in the training phase (Fig. 3b, c). In particular, during the revaluation phase, the reward associated with one of the end-boxes changes but the structure of the environment remains the same: the revaluation corresponds to a change in $\mathbf{r}$ but not $\mathbf{M}$. Therefore, the agent is able to use the DR computed during the training phase in the test phase and update its policy according to revalued reward function.

The SR is also capable of solving the latent learning task (and similar reward devaluation tasks with only a single step of actions

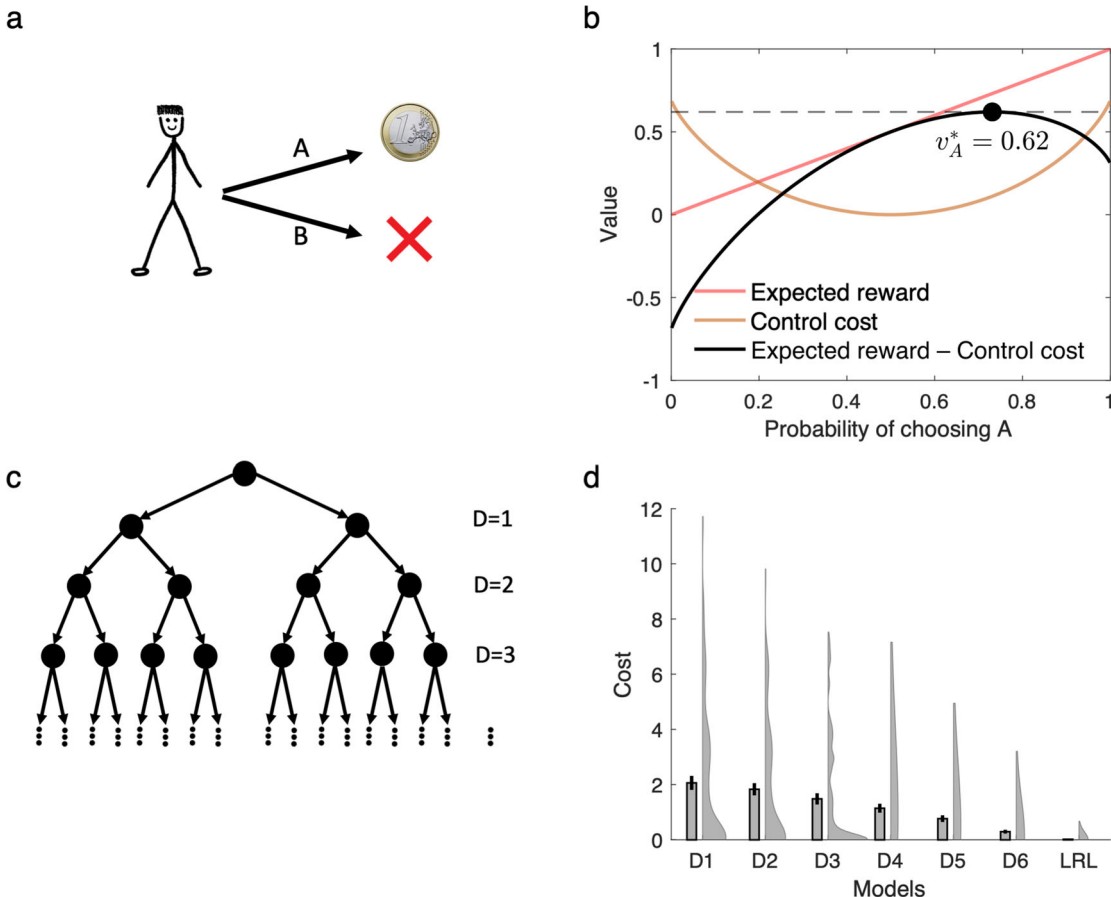

**Fig. 1 The linear RL model. a**, **b** The model optimizes the decision policy by considering the reward and the control cost, which is defined as the KL divergence between the decision policy and a default policy. Assuming an unbiased (uniform) distribution as the default policy, the optimal decision policy balances the expected reward with the control cost. Although the expected reward is maximum when probability of choosing A is close to 1 (and therefore probability of choosing B is about zero), this decision policy has maximum control cost due to its substantial deviation from the default policy. The optimal value instead maximized expected reward minus the control cost, which here occurs when probability of choosing A is 0.73. **c**, **d** The model accurately approximates optimal choice. We compared its performance on a seven-level decision tree task (with random one-step costs at each state) to six pruned model-based RL algorithms, which evaluate the task to a certain depth (D = 1,..,6; D7 is optimal; D1 is equivalent to the successor representation for the random walk policy) and use average values at the leaves. Linear RL (LRL) achieved near-optimal average costs (y-axis is additional cost relative to the optimum). Local costs of all states were randomly chosen in the range of 0–10, and simulations were repeated 100 times. Mean, standard error of the mean, and distribution of data across 100 simulations are plotted. Source data are provided as a Source Data file.

widely used in neuroscience[9]), because the SR, $\mathbf{S}^\pi$, even though learned under the original policy $\pi$, is for typical tasks good enough to compute usable new values from the new reward vector[5]. However, there are many other, structurally similar revaluation tasks—in particular, those with several stages of choices—that defeat the SR. We considered a slightly different revaluation task, which Russek et al.[5] and Momennejad et al.[6] termed "policy revaluation" that has this property. Here human subjects were first trained to navigate a three-stage sequential task leading to one of the three terminal states (Fig. 3d)[6]. The training phase was followed by a revaluation phase, in which participants experienced the terminal states with some rewards changed. In particular, a new large reward was introduced at a previously disfavored terminal state. In the final test, participants were often able to change their behavioral policy at the starting state of the task, even though they had never experienced the new terminal state contingent on their choices in the task[6].

Importantly, if the SR is learned with respect to the policy used during the training phase, then it will imply the wrong choice in the test phase (unless the successor matrix $\mathbf{S}^\pi$ is relearned or recomputed for an updated policy $\pi$), because under the original training policy, the cached successor matrix does not predict visits

to the previously low-valued state[5,17]. That is, it computes values for the top-level state (1 in Fig. 3d) under the assumption of outdated choices at the successor state 2, neglecting the fact that the new rewards, by occasioning a change in choice policy at 2 also imply a change in choice policy at 1. This task then directly probes the agent's ability to re-plan respecting the interdependence of optimal choices across states. Unlike the on-policy SR, linear RL can successfully solve this task using a DR computed for many different default policies (including a uniform default, shown here, or an optimized policy learned in the training phase), because the solution is insensitive to the default policy (Fig. 3e). (Note that because this simple example was originally designed to defeat the on-policy SR[5,17], both phases can in fact be solved by the SR for the uniform random policy. However, it is easy to construct analogous choice problems to defeat the SR for any fixed policy, including the uniform one—see also Fig. 1—so work on the SR has generally assumed that for it to be useful in planning it must be constantly updated on-policy as tasks are learned[5,6,16]).

We finally considered a different class of replanning tasks, in which the *transition* structure of the environment changes, for example, by placing a barrier onto the maze as to block the

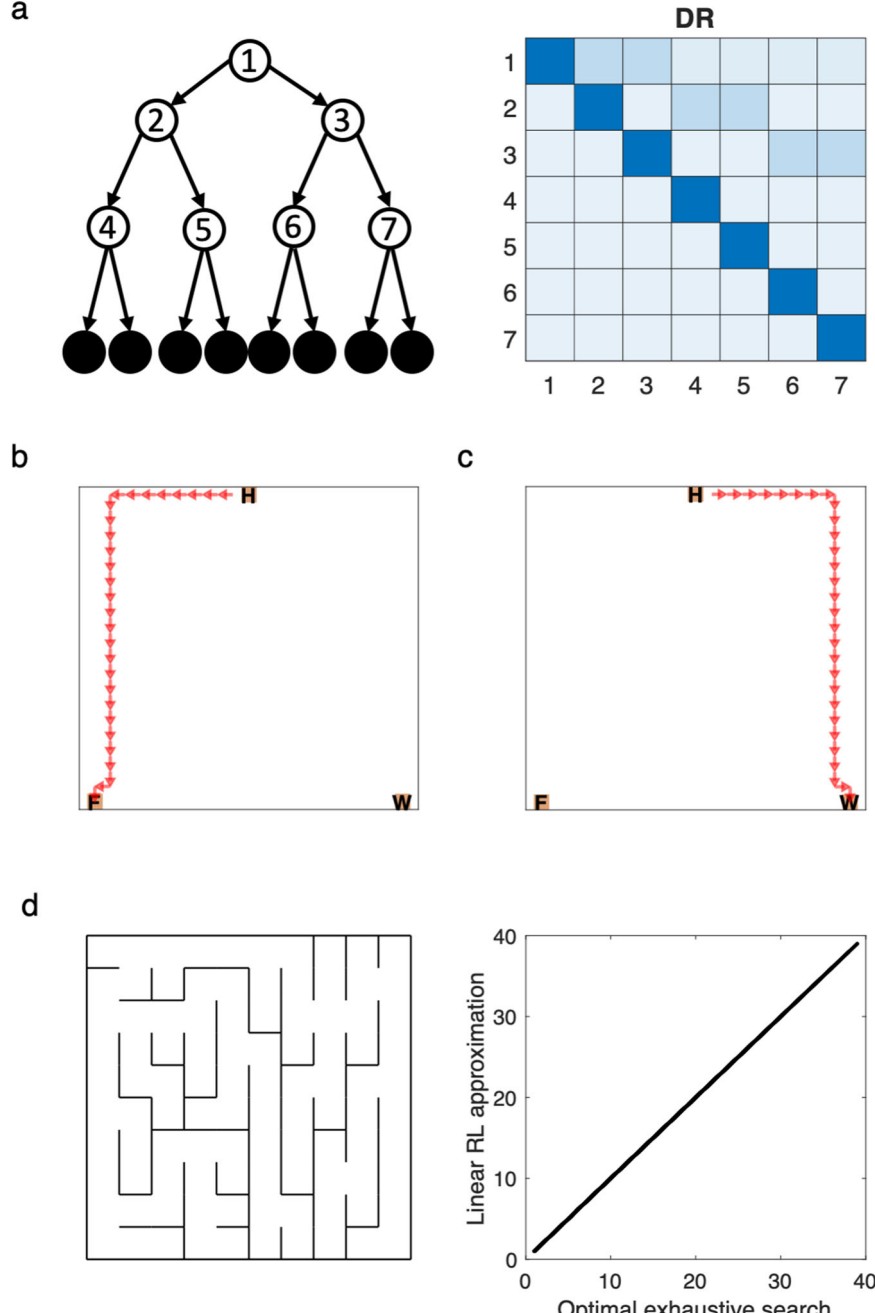

**Fig. 2 Default representation. a** The DR corresponding to a three-level decision tree task is shown. Each row of the DR represents weighted future expectancies starting from the corresponding state and following the default policy. Therefore, the DR is independent of the goals and optimized policy. **b**, **c** The optimized path for planning from home (H) to the food (F) state is computed based on the DR. The linear RL model is efficient because the same DR is sufficient for planning towards a new goal, such as the water (W) state. **d** The path between every two states in a 10 × 10 maze environment (**d**) computed by linear RL exactly matches the optimal (shortest) path computed by exhaustive search. The DR has been computed once and reused (in combination with techniques from matrix identities) to compute each optimal path.

previously preferred path[11]. These tasks pose a challenge for both the SR and DR, since the environmental transition graph is cached inside both $\mathbf{S}^{\pi}$ and $\mathbf{M}$[5,6], and these must thus be updated by relearning or recomputation in order to re-plan. However, people and animals are again often able to solve this class of revaluations[6]. We introduce an elaboration to linear RL to permit efficient solution of these tasks. In particular, we exploit matrix identities that allow us efficiently to update $\mathbf{M}$ in place to take account of local changes in the transition graph, and then re-plan as before. In particular, the updated DR, $\mathbf{M}$,

can be written as

$$\mathbf{M} = \mathbf{M}_{\text{old}} + \mathbf{M}_{\text{B}}, \qquad (5)$$

where $\mathbf{M}_{\text{B}}$ is the new term due to the barrier and it is a low-rank matrix that can be computed efficiently using $\mathbf{M}_{\text{old}}$ (see "Methods"). In fact, the rank of matrix $\mathbf{M}_{\text{B}}$ is equal to the number of states whose transition has changed. With these in place, the linear RL model can solve this task efficiently and computes the modified values and optimized policy using the old DR after updating it with simple operations (Fig. 3h).

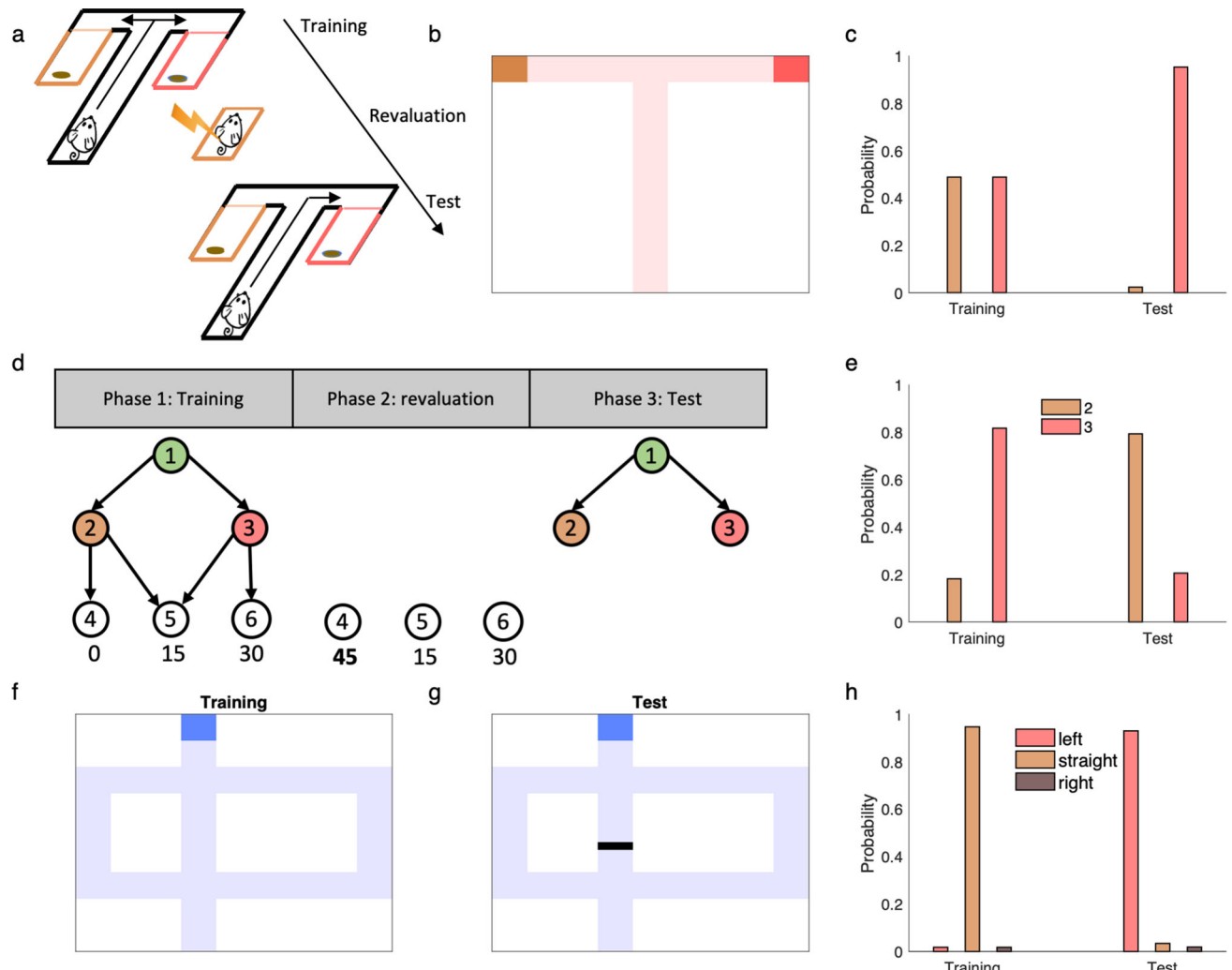

**Fig. 3 Linear RL can explain flexible replanning. a–c** Performance of linear RL on a version of Tolman's latent learning task (**a**). We simulated the model in a maze representing this task (**b**) and plotted the probability of choosing each end-box during the training and test phases. The model correctly (**c**) reallocates choices away from the devalued option. **d**, **e** Performance of linear RL in another reward revaluation task[5,6], termed policy revaluation (**d**). Choices from state 1: during the training phase, the model prefers to go to state 3 rather than state 2. Revaluation of the bottom level state reverses this preference (**e**) similar to human subjects[6]. **f–h** Performance of the model in Tolman's detour task. The structure of the environment changes in this task due to the barrier placed into the maze (**g**), which blocks the straight path. The model is able to compute the optimized policy using the old DR (following a single, inexpensive update to it) and correctly choose the left path in the test phase (**h**).

Interestingly, a similar algebraic update can also be used to update the successor matrix $\mathbf{S}^\pi$ to take account of the barrier, but this step is not in itself sufficient for replanning since the training policy $\pi$ will not be correct for the new problem. The ability of the DR to solve for the new optimal policy independent of the default policy is also required to exploit this update for efficient replanning.

**Grid fields**. The linear RL model also highlights, and suggests a resolution for, a central puzzle about the neural representation of cognitive maps or world models. It has long been argued that the brain represents a task's structure in order to support planning and flexible decision making[11]. This is straightforward for maximally local information: e.g., the one-step transition map $P(s_{t+1}|s_t, a_t)$ from Eq. (2), might plausibly be represented by connections between place fields in hippocampus, and combined with local-state reward mappings $r(s_t)$ that could be stored in hippocampal–striatal projections. But using this information for planning requires exhaustive evaluation, e.g. by replay[13], strongly suggesting a role for map-like representations of longer-

scale relationships (aggregating multiple steps) to simplify planning[19,38].

Indeed, grid cells in entorhinal cortex represent long-range (low-frequency) periodic relationships over space, and theoretical and experimental work has suggested that they play a key role in representation of the cognitive map and support navigation in both physical[24] and abstract[12,25] state spaces. However, the specific computational role of these representations in flexible planning is still unclear. A key concept is that they represent a set of basis functions for quickly building up other functions over the state space, including future value predictions like $\bar{v}^{*}$[23] and also future state occupancy predictions like the SR[2,39]. By capturing longer range relationships over the map, such basis functions could facilitate estimating or learning these functions[23]. In particular, the graph Laplacian (given by the eigenvectors of the on-policy, random walk transition matrix or, equivalently the eigenvectors of the SR for the random walk policy) generalizes Fourier analysis to an arbitrary state transition graph, and produces a set of periodic functions similar to grid fields[2,28], including potentially useful low-frequency ones. Although there

are clearly many different decompositions possible, this basic approach seems applicable to neural representations of long-run maps like the SR and DR, and potentially to compressed or regularized learning of them[23].

The puzzle with this framework is that, as mentioned repeatedly, the long-range transition map is not actually stable under changes in goals, since it depends on action choices ("max") at each step of Eq. (2): in effect, the spatial distribution of goals biases what would otherwise be a pure map of space, since those affect choice policy, which in turn affects experienced long-run location-location contingencies. Conversely, basis functions built on some fixed choice policy (like the SR for a particular $\pi$) are of limited utilty for transferring to new tasks[5,17]. Accordingly, algorithms building on these ideas in computer science (such as "representation policy iteration"[27]) iteratively update basis functions to reflect changing policies and values as each new task is learned. It has been unclear how or whether representations like this can usefully support more one-shot task transfer, as in the experiments discussed in the previous section.

As shown in the previous section, linear RL suggests a resolution for this problem, since the DR is similar to the SR but stably useful across different reward functions and resulting choice policies. In particular, the comparison between Eqs. (3) and (4) shows that the DR is a stable linear basis for the (approximate) optimal value function regardless of the reward function, but the SR is not. Accordingly, we suggest that grid cells encode an eigenvector basis for the DR, functions which are also periodic, and have grid-like properties in 2D environments (Fig. 4d). Empirically, the grid cell map is robust to some manipulations and affected by others; for our purposes here, two key classes of manipulations are those affecting which physical transitions are possible (e.g. barrier locations in space) vs. manipulations affecting which routes the animal actually tends to follow (i.e. policy). Because both the SR and DR represent relationships under the objective transition graph, both theories predict that grid fields should be affected by changes in the objective transition contingencies of the environment (e.g., barrier locations in space; though see the next section for another way to address this). This is indeed the case experimentally[29,30] (Fig. 4a–c). However, the key experimental prediction is that grid fields based on the DR can be stable under changes in the choice policy, since the default policy can be retained. Conversely the SR (and its eigenvectors) are necessarily strongly policy-dependent, so grid fields based on it should change to reflect the animal's tendency to follow particular trajectories[2]. Experimental data support the DR's prediction that grid fields are surprisingly robust to behavioral changes; for instance, grid cells are affected by walls producing a "hairpin maze" but not at all affected in rats trained to run an equivalent hairpin pattern without barriers[30] (Fig. 4a, b); grid cells are also affected by the presence or absence of a set of walls the same shape as the animal's home cage, but whether or not it is the actual home cage (which strongly affects behavioral patterns) does not change the responses[31] (Fig. 4c). Similar results have been reported in humans using functional neuroimaging[40]. A second difference between the SR and the DR is that the DR (and its eigenvectors) include information about local costs along a path; this implies the prediction that environmental features that make locomotion difficult, like rough terrain or hills, should modulate grid responses (see "Discussion").

**Border cells**. As we have already shown, one aspect of the environment that does require updating the DR if it changes is the transition structure of the environment, such as barriers. In simulating the Tolman detour task (Fig. 3f–h) we solved this problem using a matrix inversion identity, which rather than expensively recomputing or relearning the entire DR with respect to the new transition graph, expresses the new DR as the sum of the original DR plus a low-rank correction matrix reflecting, for each pair of states, the map change due to the barrier (Eq. (5)).

This operation suggests a componential way to build up spatial distance maps, such as the DR, by summing basis functions that correspond to generic components, like walls. In this case, grid cells could represent a low-rank (e.g. eigenvector) representation for a baseline map, and other cells could represent the contribution of additional environmental features. Here, we highlight the relevance and importance of this computational approach in the context of entorhinal border cells (Fig. 5a). This is another principal family of neurons in the medial entorhinal cortex that fire exclusively when the animal is close to a salient border of the environment[41], such as the wall; and are generic in the sense that they retain this tuning at least across changes in the environment's geometry. Assuming that the DR has been represented using a combination of features from a low-rank basis set, such as its eigenvectors, the columns of the matrix term for updating the DR show remarkable similarity to the border cells (Fig. 5b). This brings the border cells and grid cells under a common understanding (both as basis functions for representing the map), and helps to express this map in terms of more componential features, like walls.

In fact, our framework (Eq. (5)) implies two distinct approaches for updating the DR in light of barriers. One is to represent additional correction terms $\mathbf{M}_B$ as separate additive components, e.g. border cells. The second is to adjust the baseline map (e.g. the grid cells, $\mathbf{M}_{old}$) in place, e.g., via experiential learning or replay to incorporate the change. The latter approach implies that the geometry of the grid cells themselves would be affected by the barriers; the former that it would not be. There is some evidence that some grid cells show sensitivity to barriers and others are invariant to barriers, and that this might depend also on the extent of training in the environment[29]. Therefore, it might be the case that $\mathbf{M}_B$ is initially represented separately and later integrated into the map if the environment is stable.

**Planning in environments with stochastic transitions**. We have so far focused on environments with deterministic transitions, such as mazes, in which each action reliably selects the next state. This includes many nontrivial sequential decision tasks but excludes other stochastically controllable domains that are also relevant to biology. The assumption of fully controllable dynamics is part of what enables linear RL to work, because it allows policy optimization to occur over the continuous, differentiable space of state transition probabilities. However, it is straightforward to extend this approach to stochastic tasks by adding an additional step of approximation. First, we solve linearly for the optimal transition dynamics as though the task were fully controllable; next, choose the action selection policy that comes closest to achieving these dynamics. (This second optimization can be done in several more or less simple ways, but roughly amounts to an additional projection; see "Methods".) The question then arises to what extent this approach can account for planning in biological organisms. Here we exemplify this approach in the stochastic sequential decision task that has been most extensively studied in neuroscience, and then consider its limitations.

Consider the two-step Markov decision task, which has been widely used in psychology and neuroscience for examining the extent to which humans and animals utilize model-based vs. model-free learning[42]. Each trial of this task (Fig. 6a) consists of an action choice in each of a series of two states, followed by a

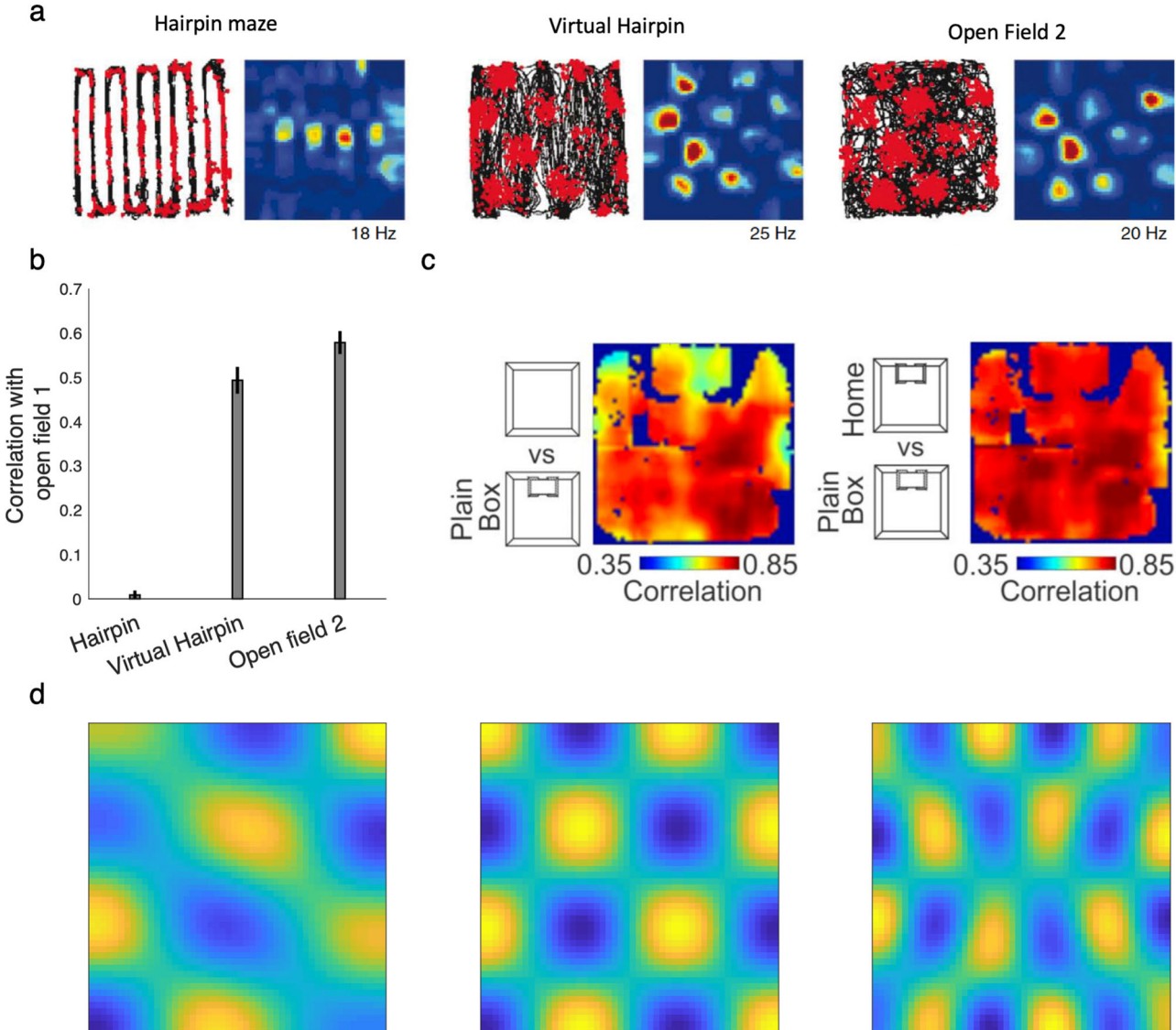

**Fig. 4 The DR as a model of grid fields. a, b** Grid fields are sensitive to the geometry of the environment, but are stable with respect to behavior (adapted with permission from Derdikman et al.[30]). Derdikman and colleagues tested grid fields in a hairpin maze formed by actual barriers, and compared them to those recorded in a "virtual" hairpin maze, in which rats were trained to show hairpin-like behavior in an open field without constraining side walls. Grid fields in the virtual hairpin differ from those in the hairpin maze but are similar to the open field. **b** This similarity is quantified by the correlation between grid fields in a baseline from an initial open field test and those from the three tasks (hairpin maze, virtual hairpin, the second control open field). This plot is adapted from Derdikman et al.[30]. Error bars are standard error of the mean. **c** Grid fields are sensitive to the presence of the home cage only insofar as it introduces new barriers in space, but not through the changes it produces in behavior (Adapted from Sanguinetti-Scheck and Brecht[31] licensed under CC BY 4.0). In particular, introducing a plain box (the same shape as the home cage) affects grid fields compared to the open field (left); but substituting the home cage for the box (right) does not further affect the grid code, although it changes behavior. The maps show the correlation between grid fields in the two scenarios. **d** All eigenvectors of the DR are independent from behavioral policies and periodic, similar to grid fields. Three example eigenvectors from a 50 × 50 maze are plotted. See Supplementary Fig. 1 for other eigenvectors. Source data are provided as a Source Data file.

terminal reward. The action choice at the first state produces a transition to one of the second-stage states, but importantly this successor is stochastic: for each first-stage action there is a common transition (with probability 0.7) and a rare one (probability 0.3). Subjects must learn to maximize the terminal reward, the chance of which is slowly diffusing from trial to trial to encourage continued policy adjustment.

The key experimental finding from this task is that humans and animals can solve it in at least a partly model-based fashion, as evidenced by the sensitivity of their switching patterns to the task's transition structure[42]. In particular, due to the symmetry of the state transitions, model-based learning predicts that if a

terminal state is rewarded (vs. not), then pursuing this reward using an updated policy implies increasing the chance of taking the same-first level action to reach the rewarded state if the experienced transition was common, but instead switching to the alternative first-level action if the experienced transition was rare. People and animals' choices display this type of sensitivity to the transition model[42]; as expected, linear RL (extended to the stochastic case) also successfully produces this pattern (Fig. 6b).

In this task—and, we conjecture, many planning tasks in stochastic domains that people can readily solve—the transition dynamics as optimized by linear RL (to transition to the better state with high probability) are similar enough to those actually

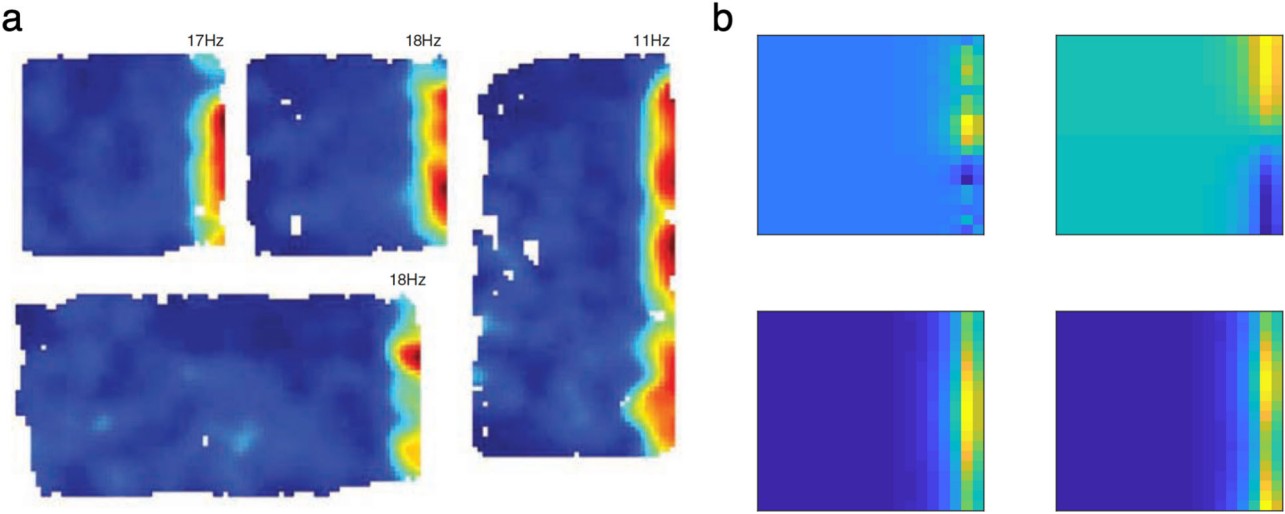

**Fig. 5 The model explains border cells. a** Rate maps for a representative border cell in different boxes; adapted from Solstad et al.[41] with permission from AAAS. **b** Columns of the matrix required to update the DR matrix to account for the wall resemble border cells. Four example columns from a 20 × 20 maze are plotted. See also Supplementary Fig. 2. Source data are provided as a Source Data file.

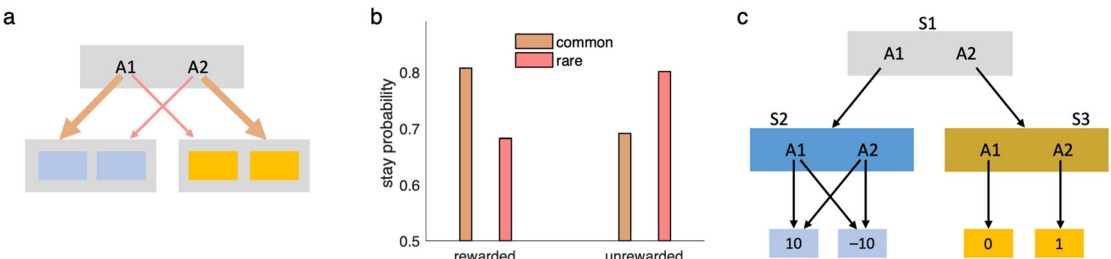

**Fig. 6 Linear RL in environments with stochastic transitions. a** The two-step Markov decision task, in which first-stage actions A1 and A2 stochastically transition to second-stage states. **b** Linear RL solves this task similar to classical model-based learning, such that the effect of reward (vs nonreward) on the subsequent first-stage choice (stay or switch, with respect to that on the current trial) depends on the type of transition (common: 70%; rare: 30%). **c** A task with stochastic transitions that linear RL fails to solve. Here, taking A1 and A2 at S1 deterministically leads to S2 and S3, respectively. However, taking either of A1 and A2 in state S2 stochastically leads to two different states with 10 and –10 reward (with 50–50% chance). Therefore, expected value of state S3 is higher than that of S2, and A2 is the optimal action in S1. Linear RL incorrectly chooses A1 in S1, however.

achievable given stochastic control (to choose the action that leads there with 70% probability). However, it is possible to construct scenarios in which this is not the case, and the approximation strategy would fail. The main issue again comes down to the interdependence of policy across states: there are cases in which ignoring action stochasticity at some state has a dramatic effect on the optimal policy at other, predecessor states. For example, in the otherwise similar Markov task of Fig. 6c, linear RL prefers A1 in S1 (while A2 is the best action on average), because it jointly optimizes the first- and second-stage transition dynamics under the assumption that the state transitions at all states are controllable. This produces an overly optimistic estimate of the value of S2 and a resulting mistake at S1. The current modeling predicts that people will either exhibit greater errors in this type of task, or instead avoid them by falling back on more costly iterative planning methods that should be measurable in longer planning times. To our knowledge, these predictions are as yet untested.

**Habits and inflexible behavior**. We have stressed the usefulness of linear RL for enabling flexible behavior. However, because this is permitted by the default policy, the model also offers a natural framework for understanding biases and inflexibilities in behavior —and phenomena of cognitive control for overcoming them—as necessary consequences of the very same computational

mechanisms. The default policy represents soft, baseline assumptions about action preferences, which (on this view) are introduced because they help efficiently though approximately to solve the problem of forecasting the trajectory of optimal future choices during planning. So far, we have simulated it as unbiased (uniform over successors), which works well because of the insensitivity of the algorithm to the default policy. However, the same insensitivity equally allows for other, non-uniform or dynamically learned default policies. In situations where action choice preferences exhibit stable regularities, it can be an even better approximation to build these in via a non-uniform default. A non-uniform default policy softly biases the model towards actions that are common under it. This aspect of the model naturally captures systematic biases in human behavior, such as habits, Stroop effects, and Pavlovian biases (next sections), and suggests a rationale for them in terms of the default policy's role in facilitating efficient planning.

In previous sections, we considered a uniform default policy that did not change in the course of decision making. Without contradicting these observations (e.g., for the relative stability of grid fields in entorhinal cortex), one can elaborate this model by assuming that the default policy might itself change gradually according to regularities in the observed transitions (i.e., in the agent's own on-policy choices). Of course, there are many ways to accomplish this; for concreteness, we use a simple error-driven

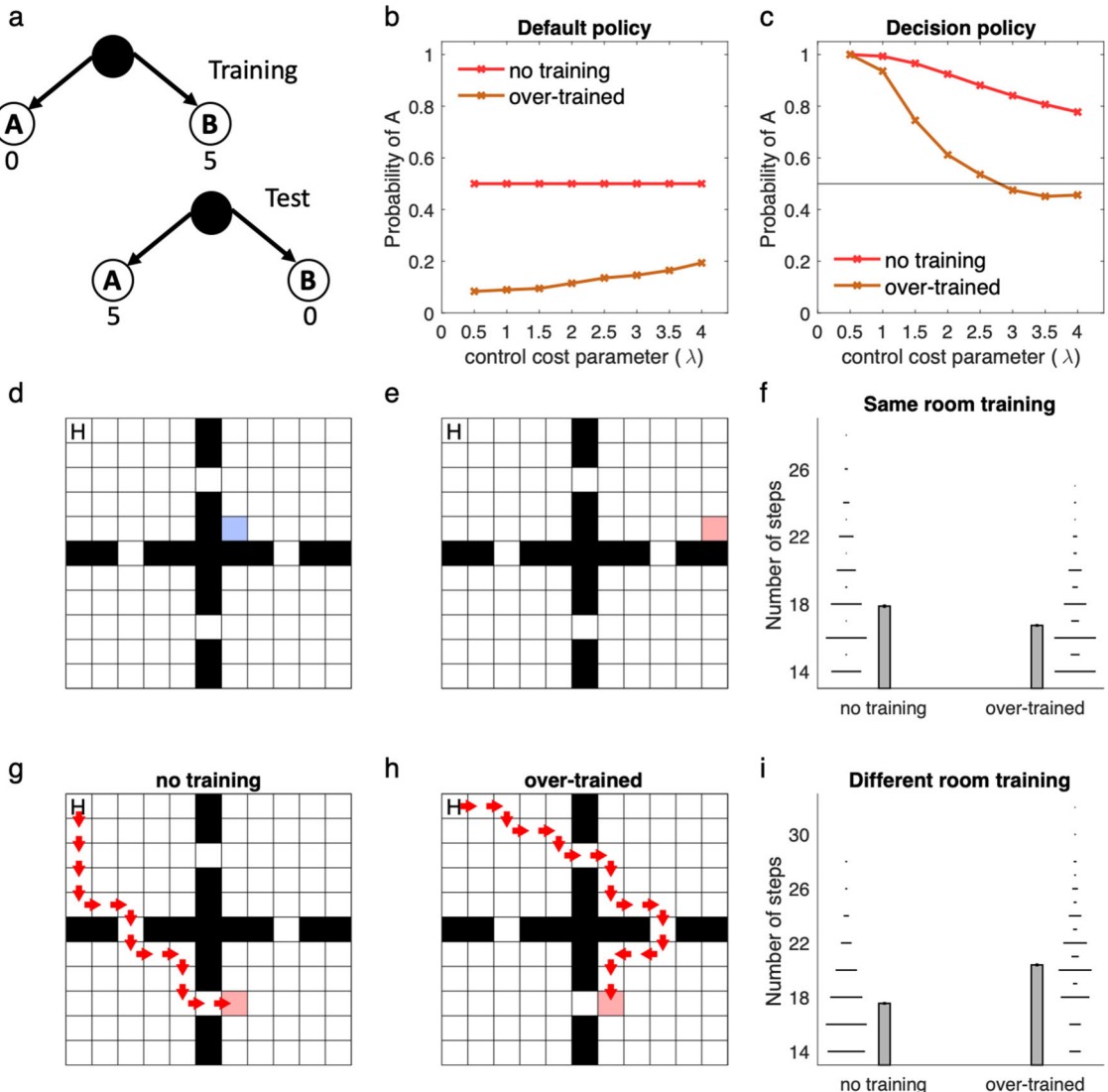

**Fig. 7 Learning the default policy results in soft habits. a–c** A simple choice task (**a**) in which the default policy has been extensively trained under conditions in which state B is rewarding. In this case, an overtrained default policy favors choice of B by default (**b**) which softly biases choice away from A even after the rewarded goal has moved in the test phase (**c**). This effect is larger when the control cost parameter, $\lambda$, is larger. This is because this parameter controls the relative weight of the control cost (for diverging from default policy; see "Methods", Eq. (6)). **d** The default policy has been trained extensively to find a goal located in the blue square. **e, f** Performance of the model with overtrained vs. uniform (i.e. no training) default policies on this task, in which the goal has been moved but it is still in the same room (**e**). The overtrained model performs better here (**f**). However, when the goal has been moved to a different room (**g–i**), the model with a uniform default policy (no training; **g**) performs better than the overtrained model, which habitually enters the room in which it has been overtrained in (**h**). Mean, standard error of the mean, and distribution of data across 1000 simulations are plotted in panels **f** and **i**. For overtraining, the model has experienced 1000 episodes of the task with step-size 0.01. Source data are provided as a Source Data file.

delta rule with a small step-size parameter (i.e. learning rate) to train the default policy (see "Methods"; simulation details for the equation). Note that there is no need to compute the DR matrix from scratch with every step of policy update. The DR can be efficiently updated from the old DR using the same matrix inversion identity used in previous sections.

In the long run, this procedure results in biases in the default policy, which then softly biases the decision policy; this produces both a higher probability of errors on individual choices (i.e., a higher chance of choosing the default action) and a resulting, more global distortion of sequential choice trajectories. Even when the step-size is small, overtraining can ultimately substantially bias the default policy toward the status quo policy. The degree to which overtraining biases the decision policy also depends on a constant parameter in the model, which scales the control cost against rewards (see "Methods" and "Discussion"; Fig. 7a–c).

Experiments with this model exemplify why a non-uniform default policy following overtraining can be relatively beneficial or harmful in some revaluation tasks. For example, when the location of a goal changes but the new location is close to the previous goal, the new policy overlaps substantially with the old one and the bias toward it is beneficial, relative to a uniform default. In Fig. 7d–i, we have simulated the model in an environment with four rooms, in which the default policy was first trained in a task in which the goal is located in the blue square (Fig. 7d). Overtraining was beneficial in a new task in which the goal was moved to a new location in the same room (Fig. 7e, f), but it was harmful in another task in which the goal was moved to a location in a different room (Fig. 7g–i). In the

latter case, the model shows a signature of habitual behavior: it prefers to enter the room that the goal was located during the course of training even though the overall resulting trajectory is suboptimal. This is because the experience obtained during training cannot be generalized to the new task: the pull of the default policy toward the pre-existing one distorts the optimum.

**Cognitive control**. Cognitive control has been defined as the ability to direct behavior toward achieving internally maintained goals and away from responses that are in some sense more automatic but not helpful in achieving those goals[33,43]. Although the basic phenomena are introspectively ubiquitous, they are also puzzling. Two classic puzzles in this area are, first, why are some behaviors favored in this way; and second, why do people treat it as costly to overcome them[44–46]? For instance, is there some rivalrous resource or energetic cost that makes some behaviors feel more difficult or effortful than others[46,47]? Such "control costs" arise naturally in the current framework, since actions are penalized if they are more unlikely under the default policy. Such deviations from default are literally charged in the objective function, in units of reward: though for computational reasons of facilitating planning, rather than energetic ones like consuming a resource. This aspect of the model is reminiscent of recent work formulating cognitive control as a decision theoretic problem, in which reward is balanced against a control-dependent cost term[46,48,49]; however, linear RL makes an explicit proposal about the functional form and nature of the cost term. (Indeed, other work in control engineering suggests alternative rationales for the same KL-divergence cost term as well; see "Discussion").

These control costs trade off in planning against the rewards for different actions, and lead (through the stochastic resulting policy) to biased patterns of errors. Figure 8a, b plots the control cost as a function of the decision policy, showing that the cost is substantially larger for choosing the action that is less likely under the default policy. For instance, action A in this simulation could be the color-naming response in the classic Stroop task, in which participants must read the name of a color that it is printed in a differently colored ink. People are faster and make fewer errors in word reading compared to color naming, presumably because the former is a more common task. For the same reason, we would expect color naming to be less likely under the default policy (as arrived at following overtraining in the organism's lifetime, as per the previous section), and incur a larger control cost to execute reliably (Fig. 8b). For any particular reward function (utility for correct and incorrect responses), this results in a larger chance of making errors for this action: a classic Stroop effect.

Furthermore, since the optimal policy in linear RL model balances the expected reward with the control cost, the model correctly predicts that these Stroop biases, although strong, are not obligatory. Instead, they can be offset by increasing the rewards for correct performance[50] (Fig. 8c, d). In other words, the prospect of reward can enhance performance even when the task is very difficult, as has been shown experimentally[50,51].

**Pavlovian-instrumental transfer**. Another example of response biases in the linear RL model arises in Pavlovian effects. Pavlovian relationships are those that arise between a stimulus and outcome (e.g. a bell and food), but not contingent on the organism's behavior. Famously, such associations when learned can trigger reflexive responses (e.g., salivation to the bell). More confusingly, such noncontingent experience can also affect later ("instrumental") choices over actions (e.g., lever-pressing for food) which are otherwise thought to be controlled by the learned association between the actions and the outcomes. This phenomenon is known as PIT. Puzzlingly, this happens even though the

Pavlovian cues are objectively irrelevant to the actions' outcomes[35,52]. PIT—in this case, associations between drug-associated cues and drugs triggering drug-seeking actions —has been argued to play a key role in the development of addiction and cue-induced relapse[53].

In a typical PIT task (Fig. 9a), animals first learn that a neutral stimulus, such as a light, predicts some rewarding outcome in a Pavlovian phase. Later, in an instrumental phase, they learn to press a lever to get the same outcome. In the final testing phase, the presentation of the conditioned stimulus biases responding toward the action for the associated reward, even though the stimulus has never been presented during instrumental phase and the stimulus is objectively irrelevant as the action produces the outcome either way (Fig. 9b). Existing RL models including the SR (and rational decision theory generally) typically fail to explain this result[54,55], instead predicting that the presence of the stimulus should not influence behavior in the test phase, because actions predict the same outcome contingencies regardless of the stimulus.

Linear RL naturally explains PIT as another example of biases arising from a learned default policy, because during the Pavlovian phase the agent should learn that the reward outcome occurs more often in the presence of the conditioned stimulus, which is reflected in the default contingencies. Therefore, during the test phase, the presentation of a conditioned stimulus elicits a default policy biased toward the corresponding outcome occurring, which favors choosing the corresponding action (Fig. 9c). Furthermore, this effect is carried by the sensory (state) aspects of the outcome, not its rewarding properties per se. In particular, since in the absence of reward, the decision policy is equal to the default policy, the theory predicts that PIT effects persist even in the absence of reward, which is consistent with experimental work showing that PIT biases survive even under reward devaluation (e.g. for food outcomes tested under satiety) (Fig. 9d, e). This finding that PIT effects reflect some sort of sensory cuing, and not reward or motivational properties of the stimulus per se, is central to the hypothesis that they underlie some phenomena in drug abuse such as cue-elicited relapse following extinction[53].

## Discussion
A central question in decision neuroscience is how the brain can store cognitive maps or internal models of task contingencies and use them to make flexible choices, and more particularly how this can be done efficiently in a way that facilitates reuse of previous computations and leverages long-run, temporally abstract predictions without compromising flexibility. To help answer this question, we identify a core issue underlying many difficulties in planning, replanning, and reuse, which is the interdependence of optimal actions across states in a sequential decision task. To solve this problem, we import from control theory[21,56] to neuroscience a computational model of decision making in the brain, called linear RL, which enables efficient (though approximate) global policy optimization by relying on soft relaxation away from default, stochastic policy expectations.

This leverages the DR, a stored, long-run predictive map of state and cost expectancies under the default policy. The DR is closely related to the SR, and inherits many of the appealing features that have generated current excitement for it as a neuroscientific model[2,5,6,57]. However, linear RL corrects serious problems that hobble the practical applicability of the SR. The DR, unlike the SR, exerts only a weak bias toward the default policy, and so delivers on the promise of a stable cognitive map[11] that can reuse substantial computation to transfer learning across contexts without sacrificing flexibility. This allows the model to

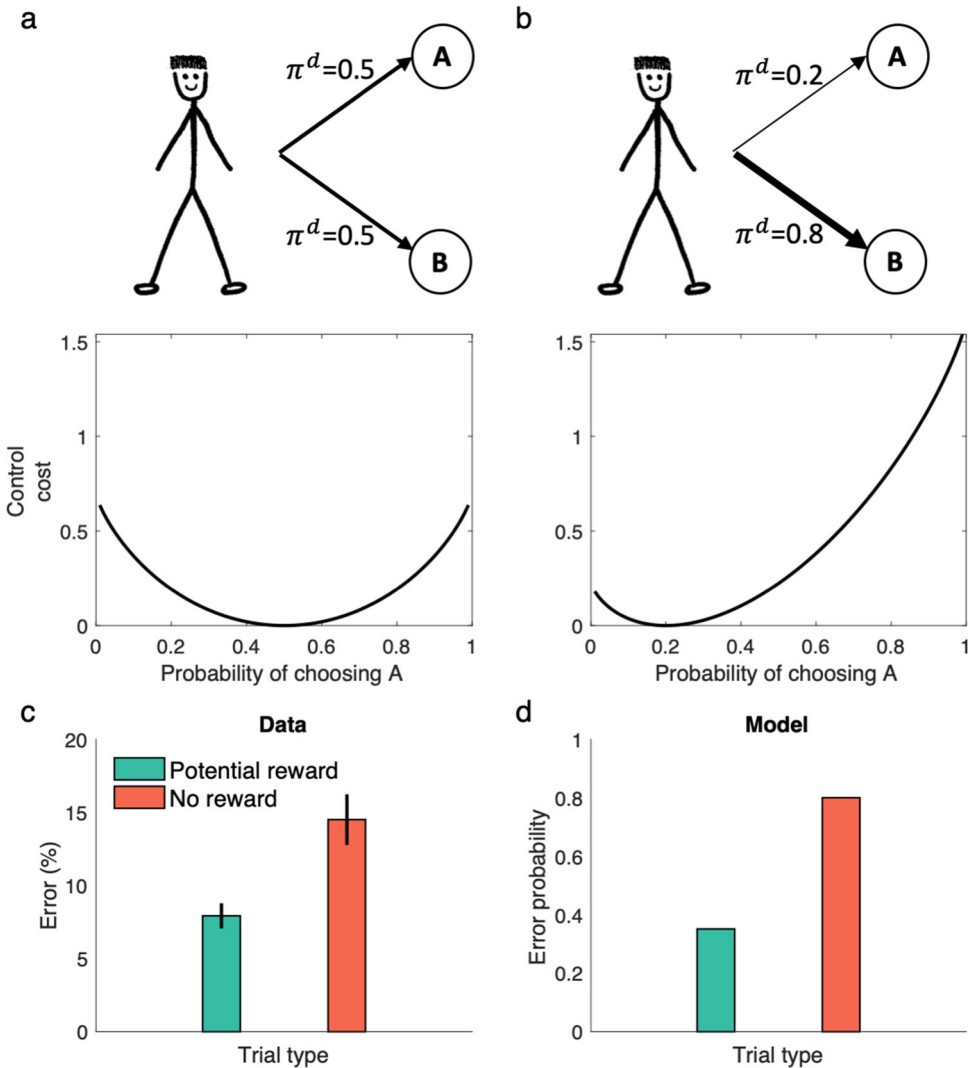

**Fig. 8 Linear RL captures prepotent actions and costs of cognitive control. a, b** The control cost is plotted as a function of the decision policy. For a uniform distribution (**a**) as the default policy, the control cost is a symmetric function of the decision policy. When the default policy is skewed toward a more likely response (**b**), the control cost is higher for reliably performing the action that is more unlikely under the default. **c** People show classical Stroop effect in a color-naming Stroop task in which the name of colors are printed in the same or different color. These errors, however, are reduced in potential reward trials, in which correct response is associated with monetary reward[51]. **d** The linear RL model shows the same behavior, because the default probability is larger for the automatic response (i.e. word reading). Promising reward reduces this effect because the agent balances expected reward against the control cost to determine the optimized policy. Data in (**c**) are adapted from Krebs et al.[51], in which mean and standard error of the mean are plotted (obtained over $n = 20$ independent samples). Source data are provided as a Source Data file.

explain animals' ability to solve reward and policy revaluation problems that otherwise would require exhaustive, biologically unrealistic model-based search. For the same reason, the model also helps to deliver on the idea that grid cells in entorhinal cortex could provide a broadly useful neural substrate for such a temporally abstract map. And the model's remaining inflexibilities—in general, soft, stochastic biases rather than hard failures—connect naturally with phenomena of cognitive control and Pavlovian biases and provide a strong theoretical framework for understanding the role of many such biases in both healthy and disordered choice.

This theory provides a unified and realistic computational framework for model-based planning in the brain and, therefore, provides a foundation for some suggestions here and much future work studying the neural substrates of different aspects of such planning. However, we should emphasize that unification at the computational level does not necessarily mean that a single neural system (e.g., entorhinal cortex) governs all these computations[58].

First, the framework encompasses many different subprocesses that have been previously associated with different brain systems (including map learning, state prediction, policy learning, value prediction, and control for overriding prepotent responses). The current framework suggests how these processes might interact, but we do not mean to imply that they are all the same thing. Furthermore, even though a particular subfunction—like map/model learning—may seem unitary in an abstract, computational sense, it may nonetheless be supported by different brain systems in different contexts, such as social vs spatial.

We motivated linear RL from a computational perspective, in which the central question is how the brain efficiently reuses previous computations for flexible replanning. Mathematically, this is enabled by introducing a control cost term, given by the dissimilarity (KL divergence) between a default policy, and the final, optimized decision policy. We argued that this penalty allows the model to explain a range of "model-based" planning and transfer phenomena, and simultaneously explain a separate

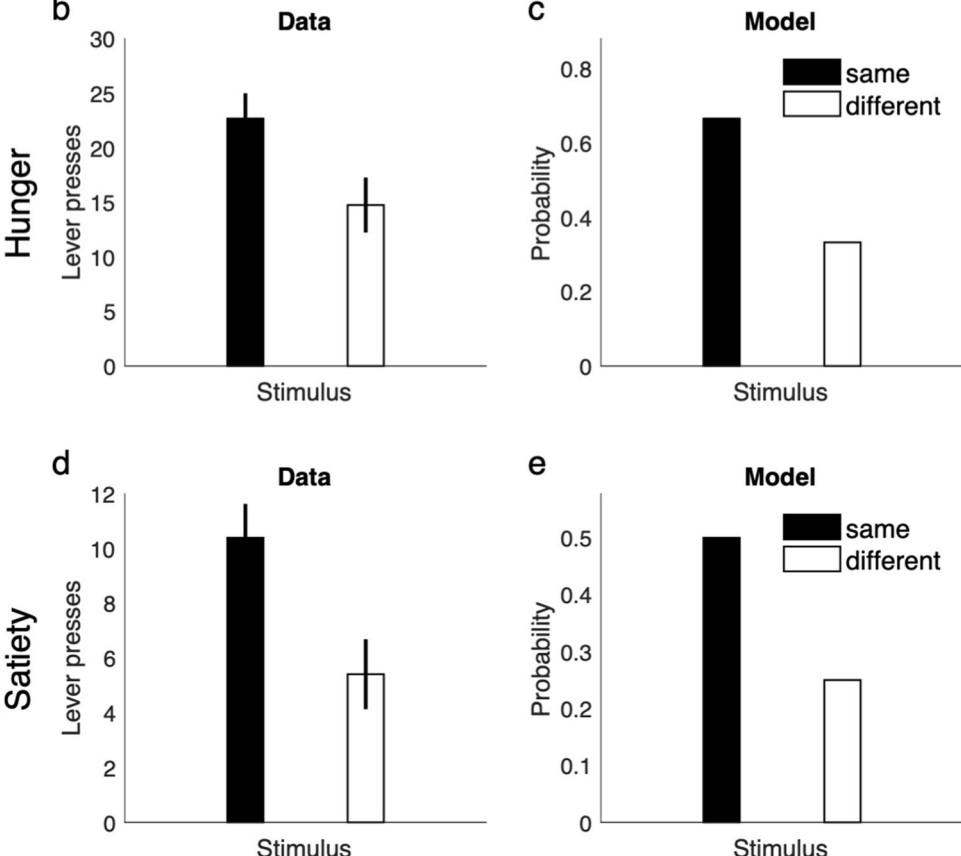

**Fig. 9 Linear RL explains Pavlovian-instrumental transfer. a** the task testing outcome-specific PIT consists of three phases: a Pavlovian training phase, an instrumental training phase, and the PIT test. Outcomes 1 and 2 are both rewarding. During PIT test, both stimuli are presented in succession, and "same" responses denote the one whose associated outcome matches that associated with the presented stimulus, e.g. Response 1 chosen following presentation of Stimulus 1. The other response is "different." **b**, **c** Data adapted from Corbit et al.[84] when rats are hungry (**b**) and simulated behavior of the model (**c**). The model learns the default policy during the Pavlovian phase, which biases performance during the PIT test. **d**, **e** Outcome-specific PIT persists even when rats are sated on both outcomes (**d**). The model shows the same behavior (**e**) because default state probabilities learned during Pavlovian training influence responses even in the absence of reward. Mean and standard error of the mean are plotted in **b** and **d**. Data in **b** and **d** are adapted from Corbit et al.[84], in which the mean and standard error of the mean are plotted (obtained over $n = 16$ independent samples). Source data are provided as a Source Data file.

set of inflexibilities in decision making, such as biased behavior, Pavlovian biases, and cognitive control (Figs. 7–9), while also providing a first-principle rationale for the "costs of control" implied by these effects. However, it is important to point out that, considered alone, these bias effects (unlike transfer) reflect only the control cost aspects of our model, and do not themselves require or exercise model-based planning. They would thus be seen, for the same reason, even in model-free algorithms for cost-

sensitive linear RL such as Todorov's Z-learning[22] and soft Q-learning[59]. Also, as discussed below, there exist alternative rationales that can motivate the same form of KL-divergence costs, where the default policy enters as a baseline expectation for efficient capacity-constrained value coding, or as a prior for Bayesian planning-as-inference. These perspectives are not necessarily mutually exclusive, but our proposal to view flexible planning as fundamental has the benefit of providing a unified

view on two important but otherwise mostly separate areas of cognitive neuroscience, i.e., flexible planning and cognitive control.

The basic planning operation in linear RL is matrix-vector multiplication, which is easily implemented in a single neural network layer. The theory offers new insights into the systems-level organization of this computation. In particular, the model realizes the promise of a representation that factors a map representing the structure of environment, separate from an enumeration of the current value of the goals in the environment. This facilitates transfer by allowing update of either of these representations while reusing the other. Previous models, like the SR, nominally exhibit this separation, but the hard policy dependence of the SR's state expectancies means that goal information, in practice, inseparably infects the map and interferes with flexible transfer[5,17].

In fact, in order to facilitate efficient planning, the linear RL model actually factors the map into three rather than two pieces, distinguishing between terminal states (representing goals), and nonterminal states (those that may be traversed on the way to goals), and dividing the map into one matrix encoding long-run interconnectivity between nonterminal states (the DR, **M**) and a second matrix representing one-step connections from nonterminal states to goals (**P**). This is a necessary restriction, in that only for this type of finite decision problem are the optimal values linearly computable. However, this classification is not inflexible, because we also introduce new techniques (based on matrix inversion lemmas) that allow dynamically changing which states are classed as goals. This allows the model (for example) to plan the best route to any arbitrary location in a maze (Fig. 2d). Representing goals as terminal states also means that the model does not directly solve problems that require figuring out how best to visit multiple goals in sequence. However, this restriction has little effect in practice because these can either be treated as a series of episodes, re-starting at each goal, or by including sub-goals within the nonterminal states, since the model does optimize aggregate cost over trajectories through nonterminal states as well.

This last point raises several interesting directions for future work. First, although there is evidence that humans choose their goal and plan towards that goal[60], there is some empirically underconstrained theoretical flexibility in specifying how a task's state space should be partitioned into terminal and nonterminal states. For the simulations here, we have tentatively adopted the principle that all discrete, punctate outcomes (like food or shock) are represented as terminal goal states with corresponding value in **r**, and the rest of the (nonterminal) states contain only costs, constant everywhere, meant to capture the cost of locomotion. But, in general, state-dependent costs can be included for nonterminal states as well. These in effect modulate the "distance" between states represented within the DR (see "Methods"). Nevertheless, this leads to the testable prediction that to whatever extent state-specific costs are accounted for within nonterminal states, they should affect hypothetical neural representations of the DR, such as grid cells. For instance, unlike for the SR, the DR predicts that by increasing locomotion cost, hills or rough terrain should increase "distance" as measured in the grid map (Supplementary Fig. 3). This aspect of the DR may be relevant for explaining recent evidence that grid cells have some subtle sensitivities to reward[61,62] which cannot be explained, as the SR-eigenvector account would predict, as secondary to changes in behavioral policy (e.g., not due to occupancy around rewarding locations[62], nor variations in trajectories or speed[61]).

Linear RL in its basic form requires one other formal restriction on tasks, compared to standard Markov decision processes as often assumed by other RL theories in theoretical neuroscience.

This is that the task is deterministically controllable. This is a good fit for many important sequential tasks, such as spatial navigation (I can reliably get from location A to location B by taking a step forward) and instrumental lever-pressing, but does not exactly map to tasks that include irreducibly stochastic state transitions. We show, however, that it is possible to address the latter class of tasks by approximating them as controllable and producing an intermediate approximate solution via linear RL. Though extremely simple, this approach can solve tasks such as two-step noisy Markov decision tasks that we and others have used to study model-based planning in humans and animals[42]. This approximation may be sufficient in practice for many realistic cases, though we also show that tasks can be constructed to defeat it (Fig. 6c). It remains to be tested how or whether people solve these cases. It may also be possible to use other forms of approximation to extend linear RL to a broader class of stochastic environments[22], but it remains for future work to explore how far this can be pushed.

We have stressed how the DR can be used for planning, and also how it embodies substantial, reusable computation (specifically, predictions of long-run future state occupancy and cost-to-go), relative to simpler, easy-to-learn map representations like the one-step state adjacency model $P(s_{t+1}|s_t)$. We have not, so far, discussed how the DR can itself be learned or computed. There are several possibilities: two inherited from previous work on the SR[5] and one newly introduced here. First, like the SR, the DR can be learned gradually by actual or replay-based sampling of the environment, using a temporal difference rule[5,16]. Second, again like the SR, the DR can be constructed from the one-step transition matrix and costs (which can themselves be learned directly by Hebbian learning) by a matrix inversion, or equivalently a sum over a series of powers of a matrix. The latter form motivates attractor methods for computing the inverse iteratively by a simple recurrent network[5,63,64].

A third possibility for learning the DR follows from the method we introduce for using matrix inversion identities to efficiently update the DR in place to add additional goals, barriers, or shortcuts (see "Methods"). This works by expressing the inverse matrix in terms of the inverses of simpler component matrices (one of which is the pre-update DR), rather than for instance by updating the transition matrix and then, expensively, re-inverting the whole thing. For instance, we used this to solve tasks, such as Tolman's detour task, in which the transition structure of the environment changes. It could also be used, state by state or barrier by barrier, as a learning rule for building up the DR from scratch.

Suggestively, this insight that the Woodbury matrix inversion identity can be used to decompose a DR map (an inverse matrix) into the sum of component maps, each associated with different sub-graphs of the transition space, offers a promising direction for a direct neural implementation for representing and constructing maps componentially: via summing basis functions, here represented by the low-rank Woodbury updates or some further function of them. This idea dovetails with—and may help to formalize and extend—the emerging idea that maps in the brain are built up by composing basis functions, such as those putatively represented in the grid cells[2,12,26,39,65]. Here, we showed that the term required to update the DR when encountering a wall remarkably resembles entorhinal border cells[41]. Therefore, our theory unifies the functional roles of entorhinal grid and border cells in planning and navigation, both as neural codes for making long-term maps that are useful for planning. We believe that this is the first step toward constructing a fully componential model of maps, although producing a border cell than can be translated and reused at different locations in compositional fashion will likely involve additional nonlinear

operations. Beyond its direct consequences, our model also opens the way for future work giving a more detailed account of different patterns of change in entorhinal maps under different environmental changes. Empirically, such changes arise both gradually and abruptly. Although we have emphasized the robustness of planning to the choice of default policy, since the DR depends on the default policy, any situations such as over-training that produce a biased default policy (Fig. 7) could ultimately and gradually lead to maps that are sensitive to the transition statistics of past behavior[66]. There are also situations in which abrupt recalculation of the DR might be necessary, for example, following substantial changes in the environment. This is broadly consistent with findings that grid fields can dramatically remap in such situations[67–69].

For making the connection between the DR and entorhinal grid fields, we followed Stachenfeld et al.[2] and used a graph Laplacian approach, in which the DR is represented using its eigen-decomposition. Although there are a number of reasons (including the parallels with entorhinal fields, and the efficiency of the Woodbury updates) to think that the brain represents maps via some decomposition similar to this, we are not committed to this specific decomposition as a mechanistic neural model. Instead, our main motivation for taking this approach was descriptive, to investigate the properties of the DR and to highlight its relationship with multiscale, periodic functions qualitatively similar to grid cells. We believe that this approach is revealing, despite its many limitations, including the fact that eigenvectors of the DR (and the SR) show a wide range of frequencies, not only hexagonally symmetric fields[65]. Notably, eigenvectors can also be learned using biologically plausible networks trained by Oja's rule[65,70], and it has been suggested that since eigen-decomposition is commonly used for compression, this approach could also be used to allow a regularized SR or DR to be learned and represented more efficiently[2]. However, this results in a loss of information if only a subset of eigenvectors is used. Nevertheless, the eigen-decomposition is by no means the only possible approach for efficient representation of the DR. In fact, the DR, at least in open fields with a fixed cost, has a redundant structure that can be exploited for efficient and exact representation, and a fully compositional account of border cells would require additional nonlinearities to account for their translation to different locations in space. This is a topic that goes beyond the scope of the current work and which we plan to pursue in future work.

Our model is based on the notion of the default policy, which is a map of expected state-to-state transition probability regardless of the current goals. Unlike previous RL models, such as the SR, linear RL does not entirely rely on the default policy and instead optimizes the decision policy around the default policy. This means that the final optimized policy is between the exact, deterministic optimized policy, and the default. The degree of this bias is controlled by a free parameter, $\lambda$, that scales the control costs relative to rewards and corresponds to the temperature in the softmax approximation to the optimization. In the limits of zero, or respectively infinite, control cost scaling, the approximation to the optimum becomes exact, or the default policy dominates completely. How should this parameter be set, and why not always take it near zero to improve the fidelity of the approximation? Linear RL works by multiplying very small numbers (future occupancies) times very large numbers (exponentiated, scaled rewards) to approximate the maximum expected value. Making this work effectively across different decision situations in the brain requires careful control of scaling to manage limits on precision (e.g. maximum spike rate and quantization, analogous to numerical precision in computers). This suggests fruitful connections (for future work) with research

on gain control and normalization[71], and rational models for choice using noisy representations[72,73].

The same tradeoff can also be understood from principles of efficient information theoretic coding[74] and from the perspective of Bayesian planning as inference[75,76]. Here, the default policy plays the role of prior over policy space and rewards play the role of the likelihood function. In this case, the decision policy is the posterior that optimally combines them[56]. Then, how much the default should influence the decision policy depends on how informative a prior it is (e.g. how reliable or uncertain it has been previously). This also suggests another distinct perspective on the default policy's role, in the model, in producing prepotent biases that can be overcome by cognitive control[43,46]. On this view, it serves to regularize behavior toward policies that have worked reliably in the past; and deviations from this baseline are presumptively costly. This perspective also provides the theoretical basis to exploit the machinery of probabilistic graphical modeling for unifying models of planning and inference in neuroscience.

Indeed, our framework can encompass many different possibilities not just for how strongly the default policy is emphasized but also how it is learned or chosen. In general, while the model provides a good approximation to the true optimal values independent of which default policy is used (so long as its cost is scaled appropriately relative to the rewards), we can also ask the converse question—which default policy should be chosen to allow for the best approximation and thereby obtain the most (actual) reward? The answer is of course, that the cost term (measuring the divergence between true and approximate $\mathbf{v}^*$) is minimized whenever the future $\pi^*$ is equal to the default $\pi^d$. Any algorithm for learning policies might be appropriate, then, for finding a $\pi^d$ that is likely to be near-optimal in the future. We exhibit one simple policy-learning algorithm (which is analogous to one habit-learning proposal from the psychological literature[77]) but other habit-learning models including model-free actor-critic learning[78] are equally applicable. A related idea has also been recently proposed in the context of a more explicitly hierarchical model of policy learning: that a default policy (and control-like charges for deviation form it) can be useful in the context of multitask learning to extract useful, reusable policies[49,79]. Separately, an analogous principle of identification of task structure that generalizes across tasks in a hierarchical generative model has also been proposed as a model of grid and place cell responses that shares some similarities with our account[12,26]. Future work remains to understand the relationship between the considerations in both of these models—which involve identifying shared structure across tasks—and ours, which are motivated instead more by efficient planning within a task.

The role of the default policy, finally, points at how the linear RL framework provides a richer, more nuanced view of habits and pathologies of decision making than previous computational theories. Although a learned default policy biases behavior, and may modulate accuracy or speed of performance, it trades off against rewards in the optimization. This give and take stands in contrast to much previous work, especially in computational psychiatry, which has often assumed a binary model of evaluation: either flexible values are computed (model-based, goal-directed) or they are not (model-free, habits). The latter, acting rather than thinking, has been taken as a model of both healthy and unhealthy habits, and especially of compulsive symptoms such as in drug abuse and obsessive compulsive disorder[80]. Although such outright stimulus-response reflexes may exist, the present framework allows for a much broader range of biases and tendencies, and may help to understand a greater range of symptomatology, such as excessive avoidance in anxiety[81], craving, and cue-induced relapse in drug abuse, and the ability to

effortfully suppress compulsive behaviors across many disorders. Finally, and relatedly, the possibility of a dynamic and situation-dependent default policy also offers a way to capture some aspects of emotion that have been resistant to RL modeling. In particular, one important aspect of emotion is its ability to elicit a pattern of congruent response tendencies, such as a greater tendency toward aggression when angry. Complementing recent work suggesting these might arise due to a hard bias on planning (via pruning context-inappropriate actions)[82], the default policy offers a clear and normative lever for influencing behavior on the basis of emotional (and other) context.

## Methods

**Model description**. In this work, we focus on MDPs with two conditions. First, we assume that there is one or a set of terminal states, $s_T$; second, we initially consider deterministic environments, such as mazes, in which there is a one-to-one map between actions and successor states (and later extend to stochastic transitions; see the section "Stochastic transitions").

The linear RL model is based on a modification to the value function for this setting[21,22], in which the agent controls the probabilistic distribution over successor states (i.e., actions) and pays an additional control cost quantified as the dissimilarity, in the form of KL divergence, between the controlled dynamics (i.e. decision policy), $\pi(.|s_t)$ and a default dynamics, $\pi^d(.|s_t)$. In particular, the objective of this MDP is to optimize a "gain" function, $g(s_t)$, defined as

$$g(s_t) = r(s_t) - \lambda \text{KL}(\pi || \pi^d) \qquad (6)$$

where $\lambda > 0$ is a constant and $\text{KL}(\pi || \pi^d)$ is the KL divergence between the two probability distributions; it is only zero if the two distributions are the same, i.e. $\pi = \pi^d$ and otherwise is positive. We also require that $\pi = 0$ if $\pi^d = 0$. Note that in the limit of zero, or respectively infinite, $\lambda$, the gain converges to pure reward (i.e. a standard MDP), or pure cost. Here, $\lambda$ scales the relative strength of control costs in units of reward (and is equivalent to rescaling the units of reward while holding the cost fixed).

It is easy then to show that the optimal value function for this new problem, $\mathbf{v}^*$, is analytically solvable[21,22] (see formal derivation below). We first define the one-step state transition matrix $\mathbf{T}$, whose $(i, j)$ element is equal to the probability of transitioning from state $i$ to state $j$ under the default policy (i.e. probability of the action under the default policy that makes $i \rightarrow j$ transition). This contains subblocks, $\mathbf{T}_{NN}$, the transition probability between nonterminal states, and $\mathbf{T}_{NT} = \mathbf{P}$, the transition probabilities from nonterminal to terminal states. Then:

$$\exp(\mathbf{v}^*/\lambda) = \mathbf{MP} \exp(\mathbf{r}/\lambda), \qquad (7)$$

where $\mathbf{v}^*$ is the vector of optimal values at nonterminal states, $\mathbf{r}$ is the vector of rewards at terminal states, and $\mathbf{M}$ is the DR matrix defined below. Note that Eq. (4) is a special case of this equation for $\lambda = 1$.

The DR matrix $\mathbf{M}$ is defined as

$$\mathbf{M} = (\text{diag}(\exp(-\mathbf{r}_N/\lambda)) - \mathbf{T}_{NN})^{-1}, \qquad (8)$$

where $\mathbf{r}_N$ is the vector of rewards at nonterminal states (which we take as a uniform cost of $-1$ in most of our simulations).

For flexibility in updating which states are viewed as goal states, it is helpful to define a second, more general version of the DR matrix, $\mathbf{D}$, defined over all states (not just nonterminal states) as

$$\mathbf{D} = (\text{diag}(\exp(-\mathbf{r}_A/\lambda)) - \mathbf{T})^{-1}, \qquad (9)$$

where $\mathbf{r}_A$ is the reward vector across all states. Note that since matrix $\mathbf{M}$ can be easily computed from $\mathbf{D}$ (in particular, $\mathbf{M}$ is a subblock of $\mathbf{D}$ corresponding to the nonterminal states only), we refer to both of them as the DR unless specified otherwise. Also note that for defining $\mathbf{D}$, we assumed, without loss of generality (since this assumption does not affect $\mathbf{M}$), that reward at terminal states are not 0.

This solution for $\mathbf{v}^*$ further implies that the policy takes the form of a weighted softmax, where the weights are given by the default policy

$$\pi(a|s_t) = \frac{\pi^d(a|s_t) \exp(v^*(s_a)/\lambda)}{\sum_{a'} \pi^d(a'|s_t) \exp(v^*(s_{a'})/\lambda)}, \qquad (10)$$

where $s_a$ is the successor state associated with action $a$. Thus, for a uniform default policy, the optimal policy is simply given by the softmax over optimal values with the temperature parameter $\lambda$. Note also that in the limit as $\lambda \rightarrow 0$, the problem becomes the classical MDP (because $g(s_t) \rightarrow r(s_t)$ in Eq. (6)) and the decision policy in Eq. (10) also reflects the optimum policy (i.e. greedy) exactly. In the limit of infinite $\lambda$, the influence of the rewards vanishes and the decision policy converges to the default policy.

**Planning toward a new goal and transfer revaluation**. Consider an environment with $\mathbf{T}_0$ and $\mathbf{D}_0$ as the transition matrix under the default policy and the associated DR, respectively. Now suppose that the agent's goal is to plan toward state $j$ (or

equivalently computing the distance between any state and $j$), i.e., we wish to add $j$ to the set of terminal states. Here, we aim to develop an efficient method to plan towards $j$ by using the cached $\mathbf{D}_0$, without re-inverting the matrix.

If we define $\mathbf{L}_0 = \text{diag}(\exp(-\mathbf{r}_A/\lambda)) - \mathbf{T}_0$ and $\mathbf{L} = \text{diag}(\exp(-\mathbf{r}_A/\lambda)) - \mathbf{T}$, then $\mathbf{L}$ and $\mathbf{L}_0$ are only different in their $j$th row (because $\mathbf{T}$ and $\mathbf{T}_0$ are only different in their $j$th row). We define $\mathbf{d}$, a vector corresponding to the difference in $j$th row of the two matrices, $\mathbf{d} = \mathbf{L}(j,:) - \mathbf{L}_0(j,:)$, and therefore, we can write:

$$\mathbf{L} = \mathbf{L}_0 + \mathbf{ed}, \qquad (11)$$

where $\mathbf{e}$ is a binary column-vector that is one only on $j$th element. Using the Woodbury matrix identity, $\mathbf{L}^{-1}$ is given by

$$\mathbf{L}^{-1} = \mathbf{L}_0^{-1} - \frac{1}{1 + \mathbf{dL}_0^{-1}\mathbf{e}} \mathbf{L}_0^{-1}\mathbf{edL}_0^{-1}, \qquad (12)$$

in which we exploited the fact that $\mathbf{d}$ and $\mathbf{e}$ are row and column vectors, respectively, and therefore $\mathbf{dL}_0^{-1}\mathbf{e}$ is a scalar. Since $\mathbf{D}_0 = \mathbf{L}_0^{-1}$ and $\mathbf{D} = \mathbf{L}^{-1}$, we obtain

$$\mathbf{D} = \mathbf{D}_0 - \frac{1}{1 + \mathbf{dm}_0} \mathbf{m}_0 \mathbf{dD}_0, \qquad (13)$$

where $\mathbf{m}_0$ is the $j$th column of $\mathbf{D}_0$.

The above equation represents an efficient, low-rank update to the DR itself. However, for the purpose of this single planning problem (e.g. if we do not intend further modifications to the matrix later), we may also further simplify the computation by focusing only on the product $\mathbf{z} = \mathbf{MP}$, which is what is needed for planning using Eq. (7) in the new environment. We find $\mathbf{z}$ in terms of an intermediate vector $\hat{\mathbf{z}} = \mathbf{D}\hat{\mathbf{P}}$, where $\hat{\mathbf{P}}$ is a subblock of $\mathbf{T}$ from all states to terminal states, in which all elements of rows corresponding to terminal states are set to 0. Therefore, $\hat{\mathbf{z}}$ is given by

$$\hat{\mathbf{z}} = \mathbf{z}_0 - \frac{1}{1 + \mathbf{dm}_0} \mathbf{m}_0 \mathbf{dz}_0, \qquad (14)$$

where

$$\mathbf{z}_0 = \mathbf{D}_0 \hat{\mathbf{P}}. \qquad (15)$$

Finally, $\mathbf{z}$ is given by the submatrix of $\hat{\mathbf{z}}$ corresponding to nonterminal rows.

It is important to note that since $\mathbf{d}$ and $\hat{\mathbf{P}}$ are very sparse, computations in Eqs. (14 and 15) are local. In fact, $\mathbf{d}$ is only nonzero on elements associated with immediate state of $j$ (and $j$th element). If we assume that there is only one terminal state (i.e. $j$), then $\hat{\mathbf{P}}$ is a vector that is nonzero on elements associated with immediate state of $j$.

The same technique can be used to update the DR or re-plan in transfer revaluation problems, such as localized changes in $\mathbf{T}_{NN}$. For example, if transition from state $j$ to $i$ has been blocked, new values for $\mathbf{D}$ and $\mathbf{z}$ can be computed efficiently using Eqs. (13) and (14), respectively. Similarly, $\mathbf{D}$ and $\mathbf{z}$ can be computed efficiently using those equations if the reward value for the nonterminal state changes. Finally, it is also possible to learn the DR matrix, transition by transition, by iteratively computing $\mathbf{D}$ for each update using $\mathbf{D}_0$ in Eq. (13).

**Border cells**. We employed a similar approach to account for border cells. Suppose that a wall has been inserted into the environment, which changes the transition matrix $\mathbf{T}_0$ to $\mathbf{T}$. Suppose $\mathbf{L}_0 = \text{diag}(\exp(-\mathbf{r}_A/\lambda)) - \mathbf{T}_0$ and $\mathbf{L} = \text{diag}(\exp(-\mathbf{r}_A/\lambda)) - \mathbf{T}$. We define matrix $\boldsymbol{\Delta}$ using rows of $\mathbf{L}_0$ and $\mathbf{L}$ corresponding to $J$:

$$\boldsymbol{\Delta} = \mathbf{L}_J - \mathbf{L}_{0J}, \qquad (16)$$

where $J$ denotes those states that their transition has been changed, $\mathbf{L}_J$ and $\mathbf{L}_{0J}$, are, respectively, submatrices associated with rows of $\mathbf{L}$ and $\mathbf{L}_0$ corresponding to $J$. Using the Woodbury matrix identity (similar to Eq. (13)), the DR associated with the new environment is given by

$$\mathbf{D} = \mathbf{D}_0 - \mathbf{B}, \qquad (17)$$

where

$$\mathbf{B} = \mathbf{D}_{0J}(\mathbf{I} + \boldsymbol{\Delta}\mathbf{D}_{0J})^{-1}\boldsymbol{\Delta}\mathbf{D}_0, \qquad (18)$$

in which matrix $\mathbf{D}_{0J}$ is the submatrix associated with columns of $\mathbf{D}_0$ corresponding to $J$, and $\mathbf{I}$ is the identity matrix. Note that although this model requires inverting of a matrix, this computation is substantially easier than inverting matrix $\mathbf{L}$, because this matrix is low dimensional. For simulating the border cells in Fig. 5, we replaced matrix $\mathbf{D}_0$ by its eigenvectors. Thus, if $\mathbf{u}$ is an eigenvector of $\mathbf{D}_0$, the corresponding column in $\mathbf{B}$, $\mathbf{b}(\mathbf{u})$ is given by

$$\mathbf{b}(\mathbf{u}) = \mathbf{D}_{0J}(\mathbf{I} + \boldsymbol{\Delta}\mathbf{D}_{0J})^{-1}\boldsymbol{\Delta}\mathbf{u}. \qquad (19)$$

**Stochastic transitions**. In deterministic environments, the default policy is equivalent to a default probabilistic mapping between states, which can be written as $\pi^d(s' = s_a|s) = \pi^d(a|s)$, where $s_a$ denotes the corresponding state (among the set of successor states of $s$) to action $a$. In environments with stochastic dynamics, however, there is no such mapping between policy and dynamics, and therefore one additional step is required to extend the linear RL framework to stochastic environments. Here, $\pi^d(s'|s)$ is defined as the default one-step transition from state

$s$ to $s'$ (it is only nonzero for successor states of $s$). We can then use the framework of linear RL to obtain the optimal transition, $u(s'|s)$, between immediate states by optimizing the gain function defined as $g(s_t) = r(s_t) - \lambda \, \mathrm{KL}(u||\pi_d)$. This is the same as Eq. (6) in which the decision policy has been replaced by $u$. Therefore, it is easy to see that optimal $u$ is given as before by Eq. (10). Now suppose that the transition model of the environment is given by $S(s'|a, s)$. The optimal $u$ can therefore be seen as the desired marginal probability distribution of the joint policy $\pi$ and the transition model, in which effects of actions are marginalized

$$u(s'|s) = \sum_a S(s'|s, a)\pi(a|s) \tag{20}$$

If we write the transition model for a given state $s$ as matrix $\mathbf{S}_s$ defined by successor states (rows) and available actions (columns), then we have $\mathbf{u}(.|s) = \mathbf{S}_s \boldsymbol{\pi}(.|s)$. We can then find $\boldsymbol{\pi}$ by minimizing the squared error between $\mathbf{u}(.|s)$ and $\mathbf{S}_s\boldsymbol{\pi}(.|s)$ under the constraint that $\boldsymbol{\pi}$ is a probability distribution. In practice, in most situations, such as the two-step task (Fig. 6a, b), this can be readily computed as $\boldsymbol{\pi}(.|s) = \mathbf{S}_s^{-1}\mathbf{u}(.|s)$. In situations in which this solution is not a distribution, an iterative optimization method (e.g. active-set) can be used. Such iterative methods converge very quickly if the rank of $\mathbf{S}_s$ is small.

**Simulation details**. We assumed a uniform default policy in all analyses presented in Figs. 1–5. In Fig. 1, the cost for all states were randomly generated in the range of 0–10 and analysis was repeated 100 times. In Fig. 2b, c, a $50 \times 50$ maze environment was considered. In Fig. 2d, e, a $10 \times 10$ maze was considered with 20 blocked states. The DR was computed in this environment with no terminal state, in which the cost for all states was 1. We used Eq. (14) to compute the shortest path using linear RL. The optimal path between every two states was computed by classic value iteration algorithm. In Fig. 3b, c, the reward of all states was –1, except the terminal states, which was +5. In the revaluation phase, the reward of the left terminal state was set to –5. In Fig. 3d, the reward of states 1, 2, and 3 is 0. In Fig. 3e, reward at all states is –1, except for the terminal state, which is +5. In Fig. 4d, a $50 \times 50$ maze was considered, the cost for all states was assumed to be 0.1. In this figure, 15th, 20th, 32th eigenvectors of the DR have been plotted. In Fig. 5b, a $20 \times 20$ maze was considered and the cost for all states was assumed to be 0.1. In this figure, 1th, 6th, 11th, and 12th eigenvectors of the DR have been considered. In Fig. 6b, the amount of reward was assumed to be 0.25. For simulating the two-step task in Fig. 6, we also assumed that there is a perseveration probability (i.e. repeating the same choice regardless of reward) of 0.75, similar to empirical values seen in our previous work[42,83]. For overtraining in Fig. 7, the model has undergone 1000 episodes of training (each until termination) and the default policy has been trained gradually according to a delta rule: if the transition is from state $S_a$ to $S_b$, and the default policy is given by $\pi^d(s_b|s_a)$, then

$\hat{\pi}^d(s_b|s_a) \leftarrow \pi^d(s_b|s_a) + \alpha(1 - \pi^d(s_b|s_a))$. The new $\hat{\pi}^d(.|s_a)$ is given by normalizing $\hat{\pi}^d(.|s_a)$. The step-size, $\alpha$, is assumed to be 0.01.

The default policy in Figs. 8 and 9 was not uniform. In Fig. 8c, the default probability for the control-demanding action assumed to be 0.2 and reward was assumed to be +2. For simulating PIT in Fig. 9, we followed experimental design of Corbit et al.[84] and assumed that the environment contains four states, in which state 1 was the choice state, states 2, 3, and 4 were associated with outcomes 1, 2 and 3, respectively. In Fig. 9c, the reward of outcome 1–3 was +5. In Fig. 9e, the reward of all states was assumed to be 0. It was also assumed that during the Pavlovian training, the default probability for Stimulus 1 → Outcome 1 and for Stimulus 2 → Outcome 2 changes from 0.33 (i.e. uniform) to 0.5.

The only parameter of linear RL is $\lambda$, which was always assumed to be 1, except for simulating the results presented in Fig. 3e (and Fig. 7, in which $\lambda$ was systematically manipulated), where we set $\lambda = 10$ to avoid overflow of the exponential due to large reward values.

**Formal derivation**. For completeness, we present derivation of Eqs. (7)–(10) based on Todorov[21,22]. By substituting the gain defined in Eq. (6) into the Bellman Eq. (2), we obtain

$$v(s_t) = r(s_t) + \max_\pi \left\{ -\lambda E_{a \sim \pi(a|s_t)} \left[ \log \frac{\pi(a|s_t)}{\pi^d(a|s_t)\exp(v(s_a)/\lambda)} \right] \right\}, \tag{21}$$

where $s_a$ denotes the corresponding state (among the set of successor states of $s_t$) to action $a$.

Note that the expectation in the Bellman equation is under the dynamics, which we have replaced it with the policy because they are equivalent here. The expression being optimized in this equation is akin to a KL divergence, except that the denominator in the argument of the log function is not normalized. Therefore, we define the normalization term $c$:

$$c = \sum_a \pi^d(a|s_t)e^{v(s_a)/\lambda}. \tag{22}$$

Note that $c$ is independent of the distribution being optimized $\pi$. By multiplying and dividing the denominator of the log by $c$, we obtain

$$v(s_t) = r(s_t) + \lambda \log c + \max_\pi \{-\lambda \mathrm{KL}(\pi(a|s_t)||\pi^d(a|s_t)e^{v(s_a)/\lambda}/c)\}, \tag{23}$$

where the maximum value of negative KL divergence is zero, which occurs only if the two distributions are equal, giving rise to Eq. (10):

$$\pi(a|s_t) = \pi^d(a|s_t)e^{v(s_a)/\lambda}/c. \tag{24}$$

Furthermore, since the KL divergence is zero, optimal values satisfy

$$v^*(s_t) = r(s_t) + \lambda \log c. \tag{25}$$

Across all states, this gives rise to a system of linear equations in the exponential space. Since at terminal states, $v(s_T) = r(s_T)$, this system can be solved analytically, which can be written in the matrix Eq. (7).

**Reporting summary**. Further information on research design is available in the Nature Research Reporting Summary linked to this article.

## Data availability
Presented data in this study are simulation data, which are available at https://github.com/payampiray/LinearRL. Source data are provided with this paper.

## Code availability
All simulations were conducted using custom code written in MATLAB v9.40 (2018a). Codes are available publicly at https://github.com/payampiray/LinearRL.

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

## Acknowledgements

We thank Tim Behrens and Jon Cohen for helpful discussions. This work was supported by grants IIS-1822571 from the National Science Foundation, part of the CRNCS program, and 61454 from the John Templeton Foundation.

## Author contributions

P.P. and N.D.D. designed the study and wrote the manuscript. P.P. performed analyses.

## Competing interests

The authors declare no competing interests.
