## [Peer Review File · Nature Communications]

REVIEWER COMMENTS

Reviewer #1 (Remarks to the Author):

Todorov's linear MDPs analysis for discrete state-spaces is used as a model of planning in biological agents. In particular, the key matrix (referred to as the "default representation") necessary to (approximately) plan via a linear readout is positioned as an improvement on the successor representation. Default representation eigendecompositions are related to grid cells and border cells. The KL-regularization on the policy objective is related to various cognitive/behavioural phenomena such as pavlovian-instrumental transfer and cognitive control costs e.g. in Stroop tasks.

Overall, the introduction of Todorov's techniques (following Kappen) to the cognitive neuroscience literature is long overdue and thus this manuscript is welcome in this regard. It touches on several topics from a distinct computational perspective. I think the manuscript would be improved with the consideration of alternative models and explanations at various points and the technical contributions could be clarified and enhanced somewhat.

Many of the simulations taken as evidence in favour of the default policy/representation are exclusively based on the KL-penalty. However, the KL-penalty is also naturally motivated by efficient coding / limited neural resources arguments. For example, is it not the case that the simulations in Figures 7, 8, and 9 would also be explained by a "capacity-constrained" RL agent without a DR representation? Such an alternative explanation has quite different implications.

Regarding the neural encoding of the DR. Eigendecomposing the DR results in periodic "eigenmaps" presumably approximately superpositions of plane waves. These are related to the firing maps of grid cells. I think more information is required regarding why the DR would be eigendecomposed. Why not just represent it in its spatial format? Is this motivated by compressing the DR? If so, what happens if I use a subset of the DR eigenvectors to plan? In particular, what is the trade-off between compression and planning error?

Minor points:

Regarding Fig 5, can the Woodbury identity not be applied also to the SR? Would this not also predict border cells via eigendecomposition?

Regarding the grid population simulation, can the DR-eigenvectors be learned?

In the introduction, it seems to me that the authors are apparently only considering planning algorithms with global optimality guarantees. If so, could this be stated explicitly.

It is stated:

"This interdependence between actions is a direct consequence of the natural definition of the objective function in this setting (i.e., the Bellman equation)"

I don't understand why the interdependence between actions is a consequence of the Bellman equation? I see it as the other way around. The Bellman equation is motivated by the interdependence problem in MDPs.

I suggest that the label "linear RL" be reconsidered. This framework is already referred to as "linear MDPs" or "linearly solvable MDPs". Furthermore, there is no reinforcement learning presented here (though there does exist an RL model associated with linear MDPs known as Z-learning).

Reviewer #2 (Remarks to the Author):

Review for "Linear reinforcement learning: Flexible reuse of computation in planning, grid fields,

and cognitive control" (Piray & Daw)

In this paper, the authors introduce a linear reinforcement learning model to account for flexible as well as inflexible behavior in decision making. The model is based on a representation of long-run state expectancies under a default policy, the default representation, and a cost associated with deviating from this default policy. The authors clearly motivate the model in the introduction before showing how it captures interesting effects in a wide range of decision problems traditionally studied in behavioral neuroscience as well phenomena related to computational psychiatry (e.g. Pavlovian biases). Furthermore, the authors relate the model to neural representations of cognitive maps, e.g. grid fields, which are thought to support flexible decision making.

I think this new computational model described in the ms makes an important contribution to decision neuroscience and more specifically to the current study of flexibilities and inflexibilities in decision making and their neural underpinnings. It also makes a couple of exciting predictions which can be tested empirically. I have the following questions and suggestions which could strengthen the ms:

p 3, paragraph 3: When the default policy and representation are first introduced in the paragraph, this could refer back to the notion of the baseline at the beginning of the paragraph to potentially further clarify the meaning of the default policy and representation.

p 3-4: When introducing the default representation, the link to the interdependent optimization problem (discussed before for the SR) should be elaborated further.

p 4, paragraph 2: The authors should add references for graded or occasional biases (Stroop effects, Pavlovian tendencies).

p 6, paragraph 2: The notion of the default policy addressing the problem of the interdependence could already be highlighted more in the introduction (e.g. p. 3, para. 3) as an explicit motivation for the model (see above).

p 6, paragraph 3: Why is it necessary to separate P and M? Maybe one could briefly elaborate on it here or refer to the elaboration of this point in the discussion.

p 7, Fig. 1d: Why is D1 equivalent to the SR?

p 9, paragraph 3: Do training phase and learning phase refer to the same phase or is there an additional training phase?

p 9, paragraph 4: Is there some quantification of the efficiency of the update to M, e.g. also with regard to the number of local changes happening?

p 10, Fig. 3c: Is the legend with the colors intended for the end-boxes (rather than the learning and test phase)? One might change the colors of the end-boxes in b so that the sides would match those in a.

p 10, Fig. 3e: Maybe change the colors of 2 and 3 to match the colors of 2 and 3 in d.

p 12, Fig. 4c: The figure caption should mention the quantification of this effect as the correlation between grid fields in the different scenarios.

p 12, Fig. 4d: The figure caption mentions that the eigenvectors are independent from behavioral policies – have the eigenvectors shown here been chosen to support this argument or are they some example eigenvectors, please clarify.

p 13, paragraph 1: Tolman detour task is shown in figure 3f-h

p 13, paragraph 1: Would the update (additional component, e.g. in case of a wall) later be

integrated into the DR or would it somehow be kept separately – and how would this then relate to grid and border cells (e.g. they continue to coexist as the original DR and the update are kept separately or are being integrated), also over the long run?

p 13, Fig. 5: While the caption currently uses both the terms “boundary” and “border” cells, it might help to stick to one of the terms throughout.

p 14, paragraph 4: Based on the model’s preference of A1 in S1 (Fig. 6c), the authors make predictions regarding human behavior in such a task (e.g. more errors). Are there already experimental findings to support these model predictions?

p 18, paragraph 2: The authors could add a reference for the Stroop effect.

p 19, paragraph 2: The authors could add references for how existing RL models fail to predict the PIT effect.

p 20: More generally regarding habits / cognitive control / PIT, how does the model here compare to the SR? Would the SR also explain e.g. the Stroop effect and PIT?

p 22, paragraph 2: The state-dependent costs for other states and the corresponding prediction regarding the neural representation of the DR are interesting discussion points. The authors could illustrate this further, e.g. the prediction, in a (supplementary) figure.

Reviewer #3 (Remarks to the Author):

This paper proposes linear reinforcement learning as a computational principle behind decision cognition and neural representations. Specifically, linear RL decomposes the optimal value function (of a modified return) into linearly combinable terms, including a ‘default’ representation (DR) prescribing a default policy that is linearly adjusted according to rewarding terminal states. It is this DR that provides the basis for explanations to various cognitive and neural phenomena.

I like this paper - it is interesting and covers plenty of ground - I think it should be published. In fact, I think this paper could be published as is, but I nevertheless give some comments below which I think would increase the impact and accessibility of this paper.

1. How have you actually learned the DR in any of these cases?

a. On p.7 you say “For linear RL, the default policy was taken as a uniform distribution over possible successor states” and on p.16 you say “we use a simple error-driven delta rule with a small step-size parameter (i.e. learning rate) to train the default policy.”

b. Which one is it? If it’s the error-driven case, then an appropriate equation would be helpful.

c. Is this learning from random exploration? Or it learned while trying to maximise reward? If so, how are you avoiding on-policy effects? This isn’t clear to me.

d. Saying SR is learned from on-policy and then comparing it the linear RL where you have either just given it the a uniform distribution or done pretraining with a uniform policy is a little unfair

e. E.g. p.9 “Importantly, this is not possible for the SR without relearning or recomputing the successor matrix SE , because under the original training policy, the cached successor matrix does not predict visits to the previously low-valued state”

f. Clearly SR can also be precomputed with a uniform policy and then it would be able to do the reward revaluation in the 2-step task

2. Border cells - surely you need to recompute the border cells should the border be placed in different locations?

a. Seems at odds with basis functions. Along these lines, what happens when an additional border is put in the middle of the room - does the same ‘border’ cell code for both walls? - this is necessary for it to be a transition basis right?

3. Some sentences are a bit funky! I put a few examples below

a. P.1 "It is thought that the brain's judicious allocation and reuse of computation underlies our ability to plan flexibly, but also failures to do so as in habits and compulsion." - I understand the meaning but don't understand the wording after the comma.

b. P.10 "The linear RL model also highlights, and resolves, a central puzzle about the neural representation of cognitive maps or world models." - It offers a resolution - I'm not sure it resolves it though!

c. P.11 "As shown in the previous section, linear RL resolves this problem, since the DR is similar to the SR but stably useful across different reward functions and resulting choice policies" - same as above

4. In numerous occasions the reader is referred to the Methods section. This is fine for the derivations, but it does obfuscate some of what's going on underneath.

a. E.g. bottom of p.9 when referring to the matrix identities - it all seems a bit magical. For me at least, it would help to include the relevant equation in the main text - it should then be easier for people to understand how your proposed role of border cells etc.

5. In Fig 1 it is claimed you compare to an exact model-based solution - I don't see this...

6. Showing all cells in the supplementary materials would be useful for the field - in particular cells for the transitions bases i.e., border cells

7. This isn't really a comment, but is for my interest's sake - do you perceive any relationship between the lambda parameter and motivation? Can this be linked to dopamine's role in motivation?

Reviewer #1 (Remarks to the Author):

Todorov's linear MDPs analysis for discrete state-spaces is used as a model of planning in biological agents. In particular, the key matrix (referred to as the "default representation") necessary to (approximately) plan via a linear readout is positioned as an improvement on the successor representation. Default representation eigendecompositions are related to grid cells and border cells. The KL-regularization on the policy objective is related to various cognitive/behavioural phenomena such as pavlovian-instrumental transfer and cognitive control costs e.g. in Stroop tasks.

Overall, the introduction of Todorov's techniques (following Kappen) to the cognitive neuroscience literature is long overdue and thus this manuscript is welcome in this regard. It touches on several topics from a distinct computational perspective. I think the manuscript would be improved with the consideration of alternative models and explanations at various points and the technical contributions could be clarified and enhanced somewhat.

We are grateful to the reviewer for their appreciation of the theory, and for their feedback. Your points helped us to improve the manuscript.

Many of the simulations taken as evidence in favour of the default policy/representation are exclusively based on the KL-penalty. However, the KL-penalty is also naturally motivated by efficient coding / limited neural resources arguments. For example, is it not the case that the simulations in Figures 7, 8, and 9 would also be explained by a "capacity-constrained" RL agent without a DR representation? Such an alternative explanation has quite different implications.

Thanks for the comment. We think it is helpful, following the reviewer's lead, to distinguish two separate aspects of the current modeling. The first is the inclusion of the KL penalty in cognitive/neural modeling, and the second is its implications specifically for map- or model-based computation.

We agree, first, that there are also model-free RL approaches like Z-learning that can also explain effects similar to those seen in Figs 7-9 for similar reasons. These particular experiments, by themselves, do not directly speak to whether the optimization is model-based (like the DR) and in general, we are open to the idea that the brain implements both MB and MF methods in different circumstances. However, we think that regardless of the algorithm, it is a contribution and a novel insight to note the implications of the KL penalty for these types of experiments.

Finally, the rest of our article, and most of the other simulations, focuses more specifically on experiments (e.g., transfer tasks, neural representations of map-like predictions) that specifically concern the implications of the linear MDP framework for questions of map-based computation, goal-directed choice, successor representations and the like. These experiments are thus more specific to these algorithmic aspects of the DR.

A related point concerns the motivation for the KL penalty, for which there are also multiple (and not necessarily mutually exclusive) rationales. We have mainly focused on justifying the penalty (and therefore positioning these cognitive biases) via computational efficiency, i.e. viewing the KL penalty as resulting from optimizing an approximation to the true objective function so as to enable fast replanning. We think this perspective is itself a contribution, and that it relates to the theme of using map-based representations to enable behavioral flexibility. But we do not mean to exclude the applicability of more

traditional arguments based on limited capacity, policy priors, planning as inference and so on.

We have endeavored to make these points clear in the revision:

In the discussion section:

We motivated linear RL from a computational perspective, in which the central question is how the brain efficiently reuses previous computations for flexible replanning. Mathematically, this is enabled by introducing a control cost term, given by the dissimilarity (KL divergence) between a default policy, and the final, optimized decision policy. We argued that this penalty allows the model to explain a range of “model-based” planning and transfer phenomena, and simultaneously explain a separate set of inflexibilities in decision making, such as biased behavior, Pavlovian biases, and cognitive control (Figures 7-9), while also providing a novel, first-principle rationale for the “costs of control” implied by these effects. However, it is important to point out that, considered alone, these bias effects (unlike transfer) reflect only the control cost aspects of our model, and do not themselves require or exercise model-based planning. They would thus be seen, for the same reason, even in model-free algorithms for cost-sensitive linear RL such as Todorov’s Z-learning²² and soft Q-learning⁵⁸. Also, as discussed below, there exist alternative rationales that can motivate the same form of KL-divergence costs, where the default policy enters as a baseline expectation for efficient capacity-constrained value coding, or as a prior for Bayesian planning-as-inference. These perspectives are not necessarily mutually exclusive, but our proposal to view flexible planning as fundamental has the benefit of providing a unified view on two important but otherwise mostly separate areas of cognitive neuroscience, i.e. flexible planning and cognitive control.

In the results section:

Cognitive control has been defined as the ability to direct behavior toward achieving internally maintained goals and away from responses that are in some sense more automatic but not helpful in achieving those goals^{33,43}. Although the basic phenomena are introspectively ubiquitous, they are also puzzling. Two classic puzzles in this area are, first, why are some behaviors favored in this way; and second, why do people treat it as costly to overcome them⁴⁴⁻⁴⁶? For instance, is there some rivalrous resource or energetic cost that makes some behaviors feel more difficult or effortful than others^{46,47}? Such “control costs” arise naturally in the current framework, since actions are penalized if they are more unlikely under the default policy. Such deviations from default are literally charged in the objective function, in units of reward: though for computational reasons of facilitating planning, rather than energetic ones like consuming a resource. **This aspect of the model is reminiscent of recent work formulating cognitive control as a decision theoretic problem, in which reward is balanced against a control-dependent cost term^{46,48,49}; however, linear RL makes an explicit proposal about the functional form and nature of the cost term. (Indeed, other work in control engineering suggests alternative rationales for the same KL-divergence cost term as well; see Discussion.)**

Regarding the neural encoding of the DR. Eigendecomposing the DR results in periodic “eigenmaps” presumably approximately superpositions of plane waves. These are related to the firing maps of grid cells. I think more information is required regarding why the DR would be eigendecomposed. Why not

just represent it in its spatial format? Is this motivated by compressing the DR? If so, what happens if I use a subset of the DR eigenvectors to plan? In particular, what is the trade-off between compression and planning error?

Thank you for this point. To be clear, we have adopted the eigen-decomposition from previous authors, notably Kim Stachenfeld, who has indeed argued (Stachenfeld et al., *Nature Neuroscience*, 2017) for spectral regularization of the SR via learning and planning using an eigen-decomposed form and retaining only a subset of vectors. Her claim is indeed that this approach could allow more efficient learning, planning etc., via compression.

The reviewer raises a number of important questions about this approach, which we think are somewhat beyond the scope of the current paper. This is because for our own part, while we think the eigen-decomposition is good enough to gain some theoretical insights about the nature of the representation, we don't actually embrace it as a practical approach and we think fully delivering on a satisfactory compressed representation of DR-like maps is an interesting (but separate) question for future work.

In the context of our paper, then, the main motivation for the eigen-decomposition of the DR was to study its properties and also compare them with those of SR in the context of grid cells. It is definitely possible to represent the DR in its original format, as suggested by the reviewer. However, we do also think that two sets of considerations point toward a representational scheme which is decomposed in some way. Computationally, the Woodbury identity suggests a useful and flexibly updatable decomposition. Empirically, the relationships (albeit imperfect) between grid and border cells and decompositions of this matrix also suggests the brain represents such maps decomposed in some way. We think the eigenvector decomposition is a useful way to explore this, but we are also aware of its limitations. We are currently working on exploiting the insights of the current modeling for efficient, compositional representation of the DR, but this is clearly future work and beyond the scope of the current manuscript.

We have now made these points clearer in the manuscript.

For making the connection between the DR and entorhinal grid fields, we followed Stachenfeld and others² and used a graph Laplacian approach, in which the DR is represented using its eigen-decomposition. Although there are a number of reasons (including the parallels with entorhinal fields, and the efficiency of the Woodbury updates) to think that the brain represents maps via some decomposition similar to this, we are not committed to this specific decomposition as a mechanistic neural model. Instead, our main motivation for taking this approach was descriptive, to investigate the properties of the DR and to highlight its relationship with multiscale, periodic functions qualitatively similar to grid cells. We believe that this approach is revealing, despite its many limitations, including the fact that eigenvectors of the DR (and the SR) show a wide range of frequencies, not only hexagonally symmetric fields⁶⁴. Notably, eigenvectors can also be learned using biologically-plausible networks trained by Oja's rule^{64,69}, and it has been suggested that since eigen-decomposition is commonly used for compression, this approach could also be used to allow a regularized SR or DR to be learned and represented more efficiently². However, this results in a loss of information if only a subset of eigenvectors is used. Nevertheless, the eigen-decomposition is by no means the only possible approach for

efficient representation of the DR. In fact, the DR, at least in open fields with a fixed cost, has a redundant structure that can be exploited for efficient and exact representation, and a fully compositional account of border cells would require additional nonlinearities to account for their translation to different locations in space. This is a topic that goes beyond the scope of the current work and which we plan to pursue in future work.

Minor points:

Regarding Fig 5, can the Woodbury identity not be applied also to the SR? Would this not also predict border cells via eigendecomposition?

The Woodbury identity is general and can be used for inverting any matrix following a low-rank perturbation. This does allow analogously updating the SR for policy to the SR for policy given an updated transition matrix, e.g. following introduction of a barrier (Figure 3f-g). However, in practice, such an update of the SR is not sufficient or even especially useful for replanning, because the barrier changes the decision policy and the SR needs to be recomputed under the new decision policy via some process of policy improvement to reflect and also find the new optimal policy. In other words, the changes in the SR that ultimately result due to introducing a barrier are not simply the low-rank update. It is the fact that the DR allows planning (relatively) independent of the default policy that makes the Woodbury update practically useful.

Regarding the grid population simulation, can the DR-eigenvectors be learned?

The eigenvectors of the DR can be learned using biologically-plausible models, such as neural networks trained with Oja's rule. We have now noted this in the discussion.

In the introduction, it seems to me that the authors are apparently only considering planning algorithms with global optimality guarantees. If so, could this be stated explicitly.

We apologize that we aren't quite sure what the reviewer has in mind here. We are certainly considering approximate planning algorithms that do not provide formal global guarantees. But much of our discussion is centered on the implications of the objective itself (ie the Bellman equation, which indeed is a global notion of optimality) rather than the guarantees actually provided by different particular algorithms that attempt (perhaps approximately) to achieve this.

It is stated:

"This interdependence between actions is a direct consequence of the natural definition of the objective function in this setting (i.e., the Bellman equation)"

I don't understand why the interdependence between actions is a consequence of the Bellman equation? I see it as the other way around. The Bellman equation is motivated by the interdependence problem in MDPs.

Thanks for this point. We agree with the reviewer that our wording is not precise and have attempted to clarify the wording. We apologize that we appear to be thinking about this in slightly different ways. On our thinking, there is nothing about MDPs per se that makes actions interdependent, since there is nothing about MDPs per se that prescribes what actions you should take in them. The interdependency follows from the *objective* of

maximizing cumulative return: i.e. the return maximizing action at some state depends on the return-maximizing action at other states. One could clearly express other objectives (eg maximizing one-step reward) that decouple actions by states. Our thinking, finally, was that the Bellman equation is the mathematical expression of the return maximizing objective function and thus the ultimate source of the dependency. But, as suggested by the reviewer, this is not strictly true, since the objective is actually to maximize the return for policy π , with respect to all π . The Bellman equation (with the max, defining optimal policy π^* implicitly) is a consequence rather than the initial statement of this objective.

We have revised the text in an attempt to reflect this:

This interdependence between actions is a consequence of the objective of maximizing cumulative expected reward in this setting and is reflected in the Bellman equation for the optimal values⁷.

I suggest that the label “linear RL” be reconsidered. This framework is already referred to as “linear MDPs” or “linearly solvable MDPs”. Furthermore, there is no reinforcement learning presented here (though there does exist an RL model associated with linear MDPs known as Z-learning).

Thanks for this suggestion. We have considered it carefully but ultimately we would like to stick to the term “linear RL.” There are two main reasons for that. First, unlike Todorov, who introduces this framework as a new class of MDPs, we have motivated this model as an approximate solution method for the classical MDP. We think the term “linear MDP” stresses the decision problem rather than the solution method. Relatedly, while our focus is not on learning in the sense of trial-and-error updating, in neuroscience (and arguably also in AI for that matter), the term “RL” and specifically “model-based RL” is commonly used to refer to solving MDPs by dynamic programming, even if (as here) the focus is on the representation and planning aspects of the problem rather than online learning per se. Thus it is natural to call this model an RL model for our neuroscience audience.

Reviewer #2 (Remarks to the Author):

In this paper, the authors introduce a linear reinforcement learning model to account for flexible as well as inflexible behavior in decision making. The model is based on a representation of long-run state expectancies under a default policy, the default representation, and a cost associated with deviating from this default policy. The authors clearly motivate the model in the introduction before showing how it captures interesting effects in a wide range of decision problems traditionally studied in behavioral neuroscience as well phenomena related to computational psychiatry (e.g. Pavlovian biases). Furthermore, the authors relate the model to neural representations of cognitive maps, e.g. grid fields, which are thought to support flexible decision making.

I think this new computational model described in the MS makes an important contribution to decision neuroscience and more specifically to the current study of flexibilities and inflexibilities in decision making and their neural underpinnings. It also makes a couple of exciting predictions which can be tested empirically. I have the following questions and suggestions which could strengthen the MS:

We are delighted by the positive evaluation of our work and thank the reviewer for very helpful suggestions. Your comments helped us to improve the manuscript considerably.

p 3, paragraph 3: When the default policy and representation are first introduced in the paragraph, this could refer back to the notion of the baseline at the beginning of the paragraph to potentially further clarify the meaning of the default policy and representation.

Thanks for this suggestion. We agree and have revised the text accordingly:

The model is based on a reformulation of the classical decision problem, which makes “soft” assumptions about the future policy (in the form of a stochastic action distribution), and introduces an additional cost for decision policies which deviate from this baseline. This can be viewed as an approximation to the classic problem, where soft, cost-dependent optimization around a baseline, which we hereafter call the default policy, stands in for exact optimization of the action at each successor state. This enables the model efficiently to deal with the interdependent optimization problem.

p 3-4: When introducing the default representation, the link to the interdependent optimization problem (discussed before for the SR) should be elaborated further.

Thanks for this suggestion. We have added the suggested link.

This can be viewed as an approximation to the classic problem, where soft, cost-dependent optimization around a baseline, which we hereafter call the default policy, stands in for exact optimization of the action at each successor state. This enables the model efficiently to deal with the interdependent optimization problem.

p 4, paragraph 2: The authors should add references for graded or occasional biases (Stroop effects, Pavlovian tendencies).

Done!

p 6, paragraph 2: The notion of the default policy addressing the problem of the interdependence could already be highlighted more in the introduction (e.g. p. 3, para. 3) as an explicit motivation for the model (see above).

Thanks for the comment. We have now revised the text to reflect the link between the default policy and the interdependent problem:

The model is based on a reformulation of the classical decision problem, which makes “soft” assumptions about the future policy (in the form of a stochastic action distribution), and introduces an additional cost for decision policies which deviate from this baseline. This can be viewed as an approximation to the classic problem, where soft, cost-dependent optimization around a baseline, which we hereafter call the default policy, stands in for exact optimization of the action at each successor state. This enables the model efficiently to deal with the interdependent optimization problem.

p 6, paragraph 3: Why is it necessary to separate P and M? Maybe one could briefly elaborate on it here or refer to the elaboration of this point in the discussion.

Thanks for the comment. We have now already elaborated on this after introducing the model:

Note that distinguishing between terminal and nonterminal states is necessary, as only for this type of finite decision problem are the optimal values linearly computable; however, this places few limits on the flexibility of the model (see Discussion).

p 7, Fig. 1d: Why is D1 equivalent to the SR?

Thanks for the comment. We have endeavored to make that clear in the revision:

). We evaluated the performance of linear RL as an approximation to exact solution by considering a difficult, 7-level decision tree task in which each state has two possible successors, a set of costs are assigned randomly at each state, and the goal is to find the cheapest path to the bottom. We conducted a series of simulations, comparing linear RL with a set of benchmarks: exact (model-based) solution, and a set of approximate model-based RL agents¹⁴ that optimally evaluate the tree up to a certain depth, then “prune” the recursion at that leaf by substituting the exact average value over the remaining subtree (Fig 1c; in the one-step case this is equivalent to the SR under the random walk policy). For linear RL, the default policy was taken as a uniform distribution over possible successor states. Except where stated explicitly, we use the same fixed uniform default policy for all simulations, so as to showcase the ability of linear RL to successfully plan without updating or relearning task-specific policy expectations, as is generally needed for the SR. Linear RL achieved near-optimal average costs (Fig 1d). **Note that the D1 model in Fig 1d is equivalent to the SR for the random walk policy (i.e. a uniform distribution over successor states), because it chooses actions using current reward plus the value of successor states computed based on a uniform policy.**

p 9, paragraph 3: Do training phase and learning phase refer to the same phase or is there an additional training phase?

Thanks for pointing this out. They refer to the same phase. We have now only used the term “training phase” in figure 3 and the corresponding results section.

p 9, paragraph 4: Is there some quantification of the efficiency of the update to M, e.g. also with regard to the number of local changes happening?

Thanks for this point. The efficiency of update to M is given by the rank of the change matrix, which is equal to the number of local changes. We have made this clear in the revision:

We finally considered a different class of replanning tasks, in which the *transition* structure of the environment changes, for example by placing a barrier onto the maze as to block the previously preferred path¹¹. These tasks pose a challenge for both the SR and DR, since the environmental transition graph is cached inside both and ^{5,6}, and these must thus be updated by relearning or recomputation in order to re-plan. However, people and animals are again often able to solve this class of revaluations⁶. We introduce a novel elaboration to linear RL to permit efficient solution of these tasks. In particular, we exploit matrix identities that allow us efficiently to update in place to take account of local changes in the transition graph, then re-plan as before. In particular, the updated DR, \tilde{D} , can be written as:

$$M = M_{\text{old}} + M_{\Delta}, \quad (1)$$

where M_{Δ} is the new term due to the barrier and it is a low-rank matrix that can be computed efficiently using M_{old} (see *Methods*). In fact, the rank of matrix M_{Δ} is equal to the number of states whose transition has changed.

p 10, Fig. 3c: Is the legend with the colors intended for the end-boxes (rather than the learning and test phase)? One might change the colors of the end-boxes in b so that the sides would match those in a.

Thanks for pointing out this problem. We have revised the figure as suggested.

p 10, Fig. 3e: Maybe change the colors of 2 and 3 to match the colors of 2 and 3 in

d. Thanks for the helpful comment. We have revised the figure as suggested.

p 12, Fig. 4c: The figure caption should mention the quantification of this effect as the correlation between grid fields in the different scenarios.

Thanks for this point. We have revised the caption as suggested.

p 12, Fig. 4d: The figure caption mentions that the eigenvectors are independent from behavioral policies – have the eigenvectors shown here been chosen to support this argument or are they some example eigenvectors, please clarify.

Thanks for the comment. All eigenvectors of the DR are independent of the behavioral policy. We have now made that clear in the manuscript.

p 13, paragraph 1: Tolman detour task is shown in figure 3f-h

Thanks for pointing out this typo.

p 13, paragraph 1: Would the update (additional component, e.g. in case of a wall) later be integrated into the DR or would it somehow be kept separately – and how would this then relate to grid and border cells (e.g. they continue to coexist as the original DR and the update are kept separately or are being integrated), also over the long run?

This is a great point. Computationally, it is possible to either integrate the additional component (i.e. M_{Δ} in Eq. 4) directly into the DR (i.e., adjusting in place the synaptic strengths that represent M_{old}); or keep both M_{old} and M_{Δ} separately, and represent the net DR via the sum of both. Notably (to the extent, in this case we associate M_{old} with the grid cells and the update with a new barrier), if the additional component is integrated into the DR, we expect to see its effects as distorting distance/geometry as measured by the grid cells. Very tentatively, as the referee also seems to suggest, we might hypothesize that map updating is initially by adding a separate component, but with much experience and incremental learning, its effects are integrated into the base map. There is some evidence that some grid cells show sensitivity to barriers and some others are invariant to barriers, and this might depend also on the extent of training in that environment (e.g. Carpenter et al, 2015). Therefore, it might be the case that M_{Δ} is initially represented separately and later integrated into the map if the environment is stable. We have now mentioned this interesting point in the revised manuscript (Results section; Border cells):

In fact, our framework (Eq. 4) implies two distinct approaches for updating the DR in light of barriers. One is to represent additional correction terms M_{Δ} as separate additive components, e.g. border cells. The second is to adjust the baseline map (e.g. the grid cells, M_{old}) in place, e.g. via experiential learning or replay to incorporate the change. The latter approach implies that the geometry of the grid cells themselves would be affected by the barriers; the former that it would not be. There is some evidence that some grid cells show sensitivity to barriers and others are invariant to barriers, and that this might depend also on the extent of training in the environment³⁰. Therefore, it might be the case that M_{Δ} is initially represented separately and later integrated into the map if the environment is stable.

p 13, Fig. 5: While the caption currently uses both the terms “boundary” and “border” cells, it might help to stick to one of the terms throughout.

Agreed. Thanks for this suggestion.

p 14, paragraph 4: Based on the model’s preference of A1 in S1 (Fig. 6c), the authors make predictions regarding human behavior in such a task (e.g. more errors). Are there already experimental findings to support these model predictions?

We believe this is a novel prediction that remains to be investigated. We have edited the text to make this clear.

The current modeling predicts that people will either exhibit greater errors in this type of task, or instead avoid them by falling back on more costly iterative planning methods that should be measurable in longer planning times. To our knowledge, these predictions are as yet untested.

p 18, paragraph 2: The authors could add a reference for the Stroop effect.

Done!

p 19, paragraph 2: The authors could add references for how existing RL models fail to predict the PIT effect.

We have added references on this point.

p 20: More generally regarding habits / cognitive control / PIT, how does the model here compare to the SR? Would the SR also explain e.g. the Stroop effect and PIT?

The linear RL produces these effects because the optimized policy is directly biased toward the default policy, a sort of Pavlovian tendency. SR does not explain Stroop effects and PIT, in both cases because it does not predict such a bias. Choice by the SR (like other RL methods) is entirely according to maximizing predicted reward V ; but although the policy it here is in some ways analogous to the default policy, the output policy is not generally biased toward it, instead it affects policy only indirectly by affecting how different states’ rewards are weighted in the value estimation. Thus, regarding the Stroop, SR does not explain the main effect because similar to other RL models, SR simply favors the more rewarding action (color naming). Similarly, for PIT, the SR simply prefers actions according to their reward contingencies. We have now explicitly noted that the SR fails to explain PIT effects for the same reason that other RL models fail to do so.

p 22, paragraph 2: The state-dependent costs for other states and the corresponding prediction regarding the neural representation of the DR are interesting discussion points. The authors could illustrate this further, e.g. the prediction, in a (supplementary) figure.

Thanks for this point. We agree that this could be helpful and therefore we have added a supplementary figure (Supp. Fig 3) to illustrate this point.

Reviewer #3 (Remarks to the Author):

This paper proposes linear reinforcement learning as a computational principle behind decision cognition and neural representations. Specifically, linear RL decomposes the optimal value function (of a modified return) into linearly combinable terms, including a ‘default’ representation (DR) prescribing a default policy that is linearly adjusted according to rewarding terminal states. It is this DR that provides the basis for explanations to various cognitive and neural phenomena.

I like this paper - it is interesting and covers plenty of ground - I think it should be published. In fact, I think this paper could be published as is, but I nevertheless give some comments below which I think would increase the impact and accessibility of this paper.

We are grateful for the reviewer’s positive evaluation of our work, strong recommendation of the manuscript, and for the insightful and helpful suggestions.

1. How have you actually learned the DR in any of these cases?

a. On p.7 you say “For linear RL, the default policy was taken as a uniform distribution over possible successor states” and on p.16 you say “we use a simple error-driven delta rule with a small step-size parameter (i.e. learning rate) to train the default policy.”

b. Which one is it? If it’s the error-driven case, then an appropriate equation would be helpful.

Thanks for this comment. We apologize that the previous draft was unclear about the details of the simulations and their rationale here.

First, in general we are not strongly committed to a particular choice of default policy. Instead, one of our main points is that linear RL is largely (and in the limit, completely) robust to it. For this reason, the first 2/3 of simulations in the paper (Figure 1-6) are all conducted assuming a fixed, uniform default policy, a choice by which we had intended to emphasize that we need not relearn the DR under each task-specific decision policy to achieve good performance, as is universally (and, as we discuss below, necessarily) assumed for SR. The remaining simulations (Figures 7-9) compare the case of nonuniform default policies, so as more explicitly to investigate the subtle biases that they imply.

We should also say that (for similar reasons) we are not committed to a particular learning rule for the DR – for concreteness, we compute the DR algebraically from the (learned or given) on-default-policy transition matrix but as we mention in the discussion it is also possible to learn it directly in various ways.

We have now made these assumptions clear in the beginning of the results section:

For linear RL, the default policy was taken as a uniform distribution over possible successor states. Except where stated explicitly, we use the same fixed uniform default policy for all

simulations, so as to showcase the ability of linear RL to successfully plan without updating or relearning task-specific policy expectations, as is generally needed for the SR.

The default policy was only learnt for results presented in Figure 7, in which we have used the error-driven delta rule. (Figures 8 and 9, in turn, use nonuniform default policies that could have been learned the same way but are instead simply assumed as we did not explicitly simulate a pretraining phase.) The learning rule equation was explicitly given in the Methods section, but now we have referred to it in the main text to avoid confusion.

In previous sections, we considered a uniform default policy that did not change in the course of decision making. Without contradicting these observations (e.g., for the relative stability of grid fields in entorhinal cortex), one can elaborate this model by assuming that the default policy might itself change gradually according to regularities in the observed transitions (i.e., in the agent's own on-policy choices). Of course, there are many ways to accomplish this; for concreteness, we use a simple error-driven delta rule with a small step-size parameter (i.e. learning rate) to train the default policy (see Methods; Simulation details for the equation).

c. Is this learning from random exploration? Or it learned while trying to maximise reward? If so, how are you avoiding on-policy effects? This isn't clear to me.

Thanks for the comment. As we stated above, the default policy was (only) learnt for results presented in Figure 7, and indeed it was learned online while maximizing reward. Thus as the reviewer intuits, it is nonuniform here; and this actually does lead to on-policy (habit-like) biases in choice, e.g. both positive (Fig 7f) and negative (7c & i) transfer relative to the baseline of a uniform default. But importantly, unlike previous models, this occurs only softly, depending on the extent of overtraining and the control cost parameter. Again, to be clear, although we show biases, the model can avoid categorical failure in transfer situations like these, because it approximates optimal value even in situations when on-(default)-policy would strictly prefer the wrong action.

d. Saying SR is learned from on-policy and then comparing it the linear RL where you have either just given it the a uniform distribution or done pretraining with a uniform policy is a little unfair.

Thanks for this comment. We don't actually see it this way. To the contrary, our view is that by for the bulk of our paper, by forcing the DR to work under a fixed, arbitrary and uninformative default policy rather than one that is continuously optimized/updated during learning, we are actually tilting the scales against it and showcasing its strength. That this is possible is exactly what differentiates it from the SR, for which this strategy is not viable. But we see why this might not have been obvious and we have endeavored to make these subtle issues clearer in the revision.

The first thing to say is that, to our knowledge, all of the previous work on the SR (including the original Dayan 1994 article, and our own studies Mommenejad et al 2017 and Russek et al 2017) has assumed it is learned on-policy and continuously updated toward improved policies as a task is learned. This is not an accident: it is because otherwise an SR-based agent would not successfully learn most tasks at all. This in turn is because the SR for policy allows computing \bar{v} (Eq. 2), but \bar{v} (in general, and especially for a generic policy like the random walk) is a very poor approximation to the optimal value function \bar{v}^* (Eq. 1). Learning a task with the SR (or other on-policy algorithms like SARSA) requires iteratively

refining the SR so that the learned value function iteratively comes to represent value under the optimal policy, while also using this iteratively to find the optimal policy.

Just to be very explicit, if you use the SR for some fixed π to compute \bar{V}^π , then choose actions that are optimal with respect to that return, this is equivalent to a single step of policy improvement for π in classic (offline) dynamic programming, like policy iteration. But policy iteration usually takes many iterations to converge, because choosing actions according to \bar{V}^π (i.e. choosing an action that is optimal at the first step, assuming you follow policy thereafter) is not in general a good standing for choosing an action that is optimal assuming you also make optimal choices later.

Oftentimes (as in the Mommenjad et al, 2017 study we revisit in Figure 3d), this point has been made using transfer tasks, in which the SR for the policy π that was optimal on some previous task is deleterious for a new task. But this problem can't be solved, in general, by simply giving the SR a fixed policy like the random walk: that, also, will be deleterious for most tasks (though, perhaps confusingly, it happens to be good enough for the particular example of Figure 3d). We demonstrate this point directly in Figure 1 of the current paper, where model D1 (the worst) is equivalent to the SR under the random walk policy. We have revised the text to make it clearer (page 8):

For linear RL, the default policy was taken as a uniform distribution over possible successor states. Except where stated explicitly, we use the same fixed uniform default policy for all simulations, so as to showcase the ability of linear RL to successfully plan without updating or relearning task-specific policy expectations, as is generally needed for the SR. Linear RL achieved near-optimal average costs (Fig 1d). **Note that the D1 model in Fig 1d is equivalent to the SR for the random walk policy (i.e. a uniform distribution over successor states), because it chooses actions using current reward plus the value of successor states computed based on a uniform policy.**

Finally, then, the main difference with the DR is that given any default policy π , we can use its DR to approximate \bar{V}^* of Eq 1 (not just \bar{V}^π), and therefore the optimal policy π^* in a single step. This is why it is not unfair to use the DR under a fixed, uniform default policy to showcase this: this approach would be a nonstarter with the SR. To reiterate, the SR is virtually useless in general if it is based on a uniform distribution (either given or slowly learnt by random walk), but linear RL works almost perfectly even under uniform distribution as the default policy (again either given or slowly learnt by random walk).

e. E.g. p.9 “Importantly, this is not possible for the SR without relearning or recomputing the successor matrix E, because under the original training policy, the cached successor matrix does not predict visits to the previously low-valued state”.

f. Clearly SR can also be precomputed with a uniform policy and then it would be able to do the reward revaluation in the 2-step task

Yes, we see now that this example is confusing because the SR under the uniform policy (though not the SR under the optimized policy) can solve this task. Honestly, in the earlier paper, we had not designed the probe to rule out uniform SR because this did not strike us as a viable hypothesis (for the reasons discussed above). However, clearly it is easy to construct similar MDPs that defeat the uniform-policy SR. For concreteness, one is shown here:

By construction, the SR for a uniform policy prefers the suboptimal action (right turn) at the top state since the value of the resulting state is higher if the choice there is random; whereas the DR for the uniform policy has the correct (left) preference, since the value is that of the better choice. Going back, then, to the example of Figure 3d, it's not the case that the DR can solve the task just because it is given the default policy (and the SR isn't); for instance, the DR can also solve the task under a default policy skewed toward the right turn at State 1 (if, for instance, the policy pretrained in the previous phase). In any case, we have stuck with the earlier example in the paper (since it corresponds to the previously published experiment), but we have added text to clarify these issues.

Importantly, if the SR is learned with respect to the policy used during the training phase, then it will imply the wrong choice in the test phase (unless the successor matrix is relearned or recomputed for an updated policy), because under the original training policy, the cached successor matrix does not predict visits to the previously low-valued state^{5,17}. That is, it computes values for the top-level state (1 in Fig 3d) under the assumption of outdated choices at the successor state (2), neglecting the fact that the new rewards, by occasioning a change in choice policy at 2 also imply a change in choice policy at 1. This task then, directly probes the agent's ability to re-plan respecting the interdependence of optimal choices across states. **Unlike the on-policy SR, linear RL can successfully solve this task using a DR computed for many different default policies (including a uniform default, shown here, or an optimized policy learned in the training phase), because the solution is insensitive to the default policy (Fig 3e). (Note that because this simple example was originally designed to defeat the on-policy SR^{5,17}, both phases can in fact be solved by the SR for the uniform random policy. However, it is easy to construct analogous choice problems to defeat the SR for any fixed policy, including the uniform one – see also Figure 1 – so work on the SR has generally assumed that for it to be useful in planning it must be constantly updated on-policy as tasks are learned^{5,6}.)**

2. Border cells - surely you need to recompute the border cells should the border be placed in different locations?

a. Seems at odds with basis functions. Along these lines, what happens when an additional border is put in the middle of the room - does the same 'border' cell code for both walls? - this is necessary for it to be a transition basis right?

We thank the reviewer for this comment. We agree with the reviewer that simple eigenvector decomposition of the DR (or the Woodbury updates to it) cannot, by itself, identify linear basis functions that are completely satisfactory for building maps: indeed, we think for reasons like the reviewer mentions, a full compositional representation of maps will involve nonlinearities beyond the scope of the current paper. For barriers and border cells, there are actually two issues, one of which the reviewer mentions: a border cell for a barrier at a different location has analogous structure but is shifted within the matrix (which is not a linear operation, i.e. it is a different "basis function," albeit one that

is clearly analogous). Also, the Woodbury expressions imply that the effect of two barriers on the DR is not just the sum of each barrier's contribution if it had been included alone: there is a further interaction, another violation of linearity. Having said that, we believe that the same border cell in our model (i.e. a code that is consistent with the $b(u)$ in the equation given in Methods, section "Border cells") captures the form of low-rank perturbation due to a single wall regardless of the position of the wall, up to a translation/shift.

In all, we did not mean to imply that the current model has solved the problem of basis functions for building maps. Our main point is that the Woodbury matrix identity enables us to move in that direction in the future, and also that together with the eigenvector analysis this is good enough to understand the role of border cells as corrections to a DR-like map, even though generatively producing a master "border cell" that can be translated and reused at different locations in compositional fashion will involve additional nonlinear operations. We have revised the discussion to make it clear that this is a crucial point for future work.

Suggestively, this insight that the Woodbury matrix inversion identity can be used to decompose a DR map (an inverse matrix) into the sum of component maps, each associated with different sub-graphs of the transition space, offers a promising direction for a direct neural implementation for representing and constructing maps componentially: via summing basis functions, here represented by the low-rank Woodbury updates or some further function of them. This idea dovetails with – and may help to formalize and extend – the emerging idea that maps in the brain are built up by composing basis functions, such as those putatively represented in the grid cells^{2,12,26,39,66}. Here, we showed that the term required to update the DR when encountering a wall remarkably resembles entorhinal border cells⁴¹. Therefore, our theory unifies the functional roles of entorhinal grid and border cells in planning and navigation, both as neural codes for making long-term maps that are useful for planning. We believe that this is the first step toward constructing a fully componential model of maps, which should be thoroughly studied in the future. Beyond its direct consequences, our model also opens the way for future work giving a more detailed account of different patterns of change in entorhinal maps under different environmental changes. Empirically, such changes arise both gradually and abruptly. Although we have emphasized the robustness of planning to the choice of default policy, since the DR depends on the default policy, any situations such as overtraining that produce a biased default policy (Fig 7), could ultimately and gradually lead to maps that are sensitive to the transition statistics of past behavior⁶⁷. There are also situations in which abrupt recalculation of the DR might be necessary, for example following substantial changes in the environment. This is broadly consistent with findings that grid fields can dramatically remap in such situations^{68–70}.

3. Some sentences are a bit funky! I put a few examples below

a. P.1 "It is thought that the brain's judicious allocation and reuse of computation underlies our ability to plan flexibly, but also failures to do so as in habits and compulsion." - I understand the meaning but don't understand the wording after the comma.

b. P.10 "The linear RL model also highlights, and resolves, a central puzzle about the neural representation of cognitive maps or world models." - It offers a resolution - I'm not sure it resolves it though!

c. P.11 “As shown in the previous section, linear RL resolves this problem, since the DR is similar to the SR but stably useful across different reward functions and resulting choice policies” - same as above

We agree and have adjusted these passages, thank you.

4. In numerous occasions the reader is referred to the Methods section. This is fine for the derivations, but it does obfuscate some of what’s going on underneath.

a. E.g. bottom of p.9 when referring to the matrix identities - it all seems a bit magical. For me at least, it would help to include the relevant equation in the main text - it should then be easier for people to understand how your proposed role of border cells etc.

We agree with the reviewer that it is better to present that result in a formal form in the main text and we have now revised the text accordingly.

We introduce a novel elaboration to linear RL to permit efficient solution of these tasks. In particular, we exploit matrix identities that allow us efficiently to update M in place to take account of local changes in the transition graph, then re-plan as before. In particular, the updated DR, M , can be written as:

$$M = M_{\text{old}} + M_{\Delta}, \quad (2)$$

where M_{Δ} is the new term due to the barrier and it is a low-rank matrix that can be computed efficiently using M_{old} (see *Methods*). In fact, the rank of matrix M_{Δ} is equal to the number of states whose transition has changed.

5. In Fig 1 it is claimed you compare to an exact model-based solution - I don’t see this...

Thanks for this point. We have plotted relative cost in Fig 1d, in which the baseline (i.e. 0) is the cost of exact model-based algorithm. We have made that clear in the revision.

6. Showing all cells in the supplementary materials would be useful for the field - in particular cells for the transitions bases i.e., border cells

We agree. As suggested, we have now included two supplementary figures regarding both grid and border cells.

7. This isn’t really a comment, but is for my interest’s sake - do you perceive any relationship between the lambda parameter and motivation? Can this be linked to dopamine’s role in motivation?

This is an interesting idea; we have not thought about the lambda parameter in this way. As we mention in the MS, there are multiple interpretations of this parameter, but these all basically involve balancing the strength of reward maximization against other considerations (e.g. prior policy information or representational costs of decision variables). In this sense, it has some resonance with motivation, and in turn potentially might, quite hypothetically, be implemented by scaling up or down reward information as reported by dopamine, e.g. the gain of phasic responses or their contrast against background / tonic dopamine.

REVIEWERS' COMMENTS

Reviewer #1 (Remarks to the Author):

Thank you for fully addressing my comments. I have no further queries and would be happy to see this interesting manuscript accepted and published.

Reviewer #2 (Remarks to the Author):

I am very happy about the authors' responses to my questions and the incorporation of my suggestions.

Reviewer #3 (Remarks to the Author):

I thank the authors for their responses.

The authors can feel free to ignore this, but I feel it would be useful, and less misleading, to explicitly mention that the basis representations would need to be different for walls/barriers places at different locations in the environment.

Otherwise, this is a good paper and should be published.

Reviewers' comments

Reviewer #1 (Remarks to the Author):

Thank you for fully addressing my comments. I have no further queries and would be happy to see this interesting manuscript accepted and published.

Reviewer #2 (Remarks to the Author):

I am very happy about the authors' responses to my questions and the incorporation of my suggestions.

Reviewer #3 (Remarks to the Author):

I thank the authors for their responses.

The authors can feel free to ignore this, but I feel it would be useful, and less misleading, to explicitly mention that the basis representations would need to be different for walls/barriers places at different locations in the environment.

Otherwise, this is a good paper and should be published.

We thank the reviewer for raising this last point. We have revised the text to address this concern:

Suggestively, this insight that the Woodbury matrix inversion identity can be used to decompose a DR map (an inverse matrix) into the sum of component maps, each associated with different sub-graphs of the transition space, offers a promising direction for a direct neural implementation for representing and constructing maps componentially: via summing basis functions, here represented by the low-rank Woodbury updates or some further function of them. This idea dovetails with – and may help to formalize and extend – the emerging idea that maps in the brain are built up by composing basis functions, such as those putatively represented in the grid cells^{2,12,26,39,66}. Here, we showed that the term required to update the DR when encountering a wall remarkably resembles entorhinal border cells⁴¹. Therefore, our theory unifies the functional roles of entorhinal grid and border cells in planning and navigation, both as neural codes for making long-term maps that are useful for planning. **We believe that this is the first step toward constructing a fully componential model of maps, although producing a border cell model than can be translated and reused at different locations in compositional fashion will likely involve additional nonlinear operations.**